# WithAnyone: Toward Controllable and ID Consistent Image Generation

**Hengyuan Xu**[1,2]    **Wei Cheng**[2,†]    **Peng Xing**[2]    **Yixiao Fang**[2]    **Shuhan Wu**[2]    **Rui Wang**[2]
**Xianfang Zeng**[2]    **Daxin Jiang**[2]    **Gang Yu**[2,‡]    **Xingjun Ma**[1,‡]    **Yu-Gang Jiang**[1]

[1] Fudan University    [2] StepFun

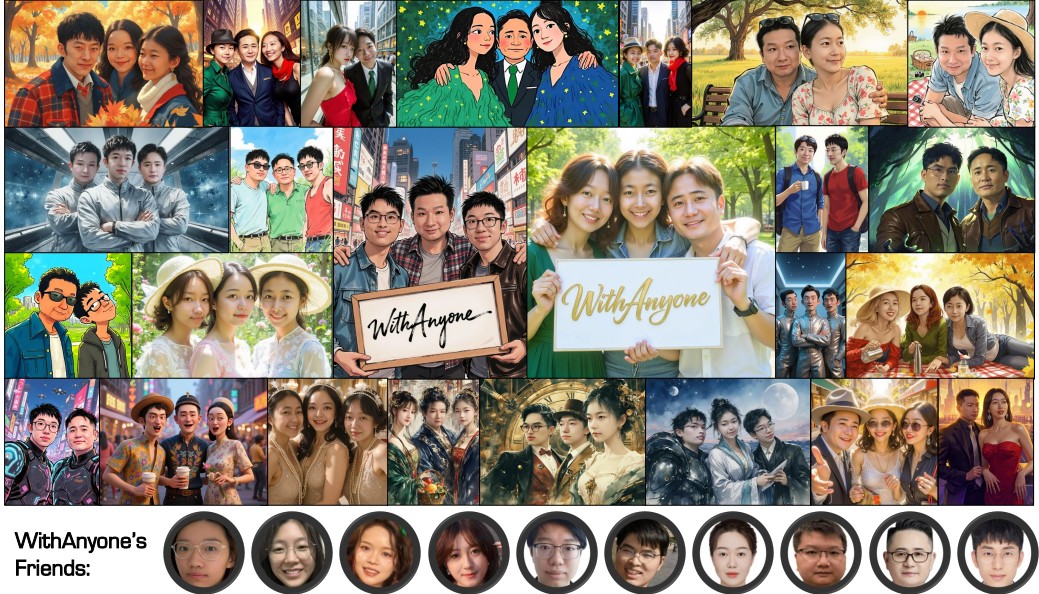

Figure 1: **Showcases of WithAnyone.** WithAnyone is capable of generating high-quality, controllable, and ID-consistent images by leveraging ID-contrastive training on the proposed **MultiID-2M** dataset. IDs above are authors' and authors' friends'.

## Abstract

Identity-consistent (ID-consistent) generation has become an important focus in text-to-image research, with recent models achieving notable success in producing images aligned with a reference identity. Yet, the scarcity of large-scale paired datasets—containing multiple images of the same individual—forces most approaches to adopt reconstruction-based training. This reliance often leads to a failure mode we term *copy-paste*, where the model directly replicates the reference face rather than preserving identity across natural variations in pose, expression, or lighting. Such over-similarity undermines controllability and limits the expressive power of generation. To address these limitations, we (1) construct a large-scale paired dataset **MultiID-2M** tailored for multi-person scenarios, providing diverse references for each identity; (2) introduce a benchmark that quantifies both copy-paste artifacts and the trade-off between identity fidelity and variation; and (3) propose a novel training paradigm with a contrastive identity loss that leverages paired data to balance fidelity with diversity. These contributions culminate in **WithAnyone**, a diffusion-based model that effectively mitigates copy-paste while preserving high identity similarity. Extensive experiments—both qualitative and quantitative—demonstrate that WithAnyone substantially reduces copy-paste artifacts, improves controllability over pose and expression, and maintains strong perceptual quality. User studies further validate that our method achieves high identity fidelity while enabling expressive, controllable generation. Our project is fully open-sourced at HTTPS://DOBY-XU.GITHUB.IO/WITHANYONE/.

# 1 INTRODUCTION

With the rapid progress of generative AI, controllable image generation via reference images or image prompting (Ruiz et al., 2023; Hertz et al., 2022; Zhang et al., 2023a; Xiao & Fu, 2024; Hu et al., 2025b; Wu et al., 2024a) and identity-consistent (ID-consistent) generation (Ye et al., 2023; Guo et al., 2024; Wang et al., 2024c; Jiang et al., 2025a; Cheng et al., 2025; Zhang et al., 2025; He et al., 2024) have achieved remarkable advances: modern models can synthesize portraits that closely match the provided individual. Recent efforts (Cheng et al., 2025; Chen et al., 2025) push resemblance toward near-perfect reproduction. While pursuing higher similarity seems natural, beyond a certain point, excessive fidelity becomes counterproductive.

In real photographs of the same person, identity similarity varies substantially due to natural changes in pose, expression, makeup, and illumination (Fig. 2). By contrast, many generative models adhere to the reference image far more rigidly than this natural range of variation. Although such over-optimization may seem beneficial, it suppresses legitimate variation, reducing controllability and limiting practical usability. We term this failure mode the **copy-paste artifact**: rather than synthesizing an identity in a flexible, controllable manner, the model effectively copies the reference image into the output (see Fig. 2). In this work, we formalize this artifact, develop metrics to quantify it, and propose a novel training strategy to mitigate it.

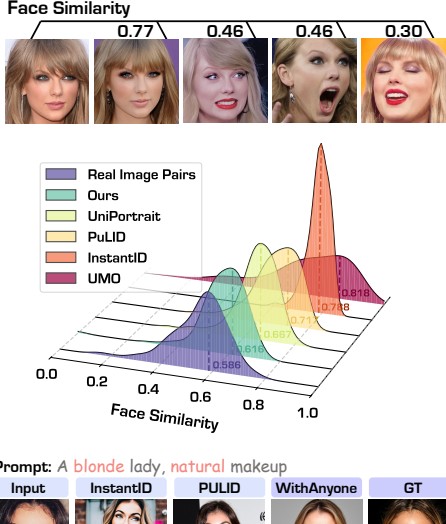

Figure 2: **Our Observation**. Natural variations, such as head pose, expression, and makeup, may cause more face similarity decrease than expected. Copying reference image limits models' ability to respond to expression and makeup adjustment prompts.

Mitigating copy-paste artifacts is fundamentally constrained by the lack of suitable training data. While numerous large-scale face datasets exist (Liu et al., 2015; Stacchio et al., 2020; Chu et al., 2024; Zhang et al., 2015; Jiang et al., 2025b; Zhong et al., 2018; Wang et al., 2025), they remain ill-suited for controllable multi-identity generation. Critically, few datasets provide paired references for each identity—multiple images of the same person across diverse expressions, poses, hairstyles, and viewpoints. As a result, most prior work resorts to single-person, reconstruction-based training (Guo et al., 2024; Wang et al., 2024c), where the reference and target coincide. This setup inherently promotes copying and exacerbates copy-paste artifacts. Constructing datasets with multiple references per identity, particularly in group photos, and developing methods to effectively exploit such data remain open challenges.

In this work, we introduce a large-scale open-source Multi-ID dataset, **MultiID-2M**, together with a comprehensive benchmark, **MultiID-Bench**, designed for intrinsic evaluation of multi-identity image generation. MultiID-2M contains 500k group photos featuring 1–5 recognizable celebrities. For each celebrity, hundreds of individual images are provided as paired references, covering diverse expressions, hairstyles, and viewing angles. In addition, 1.5M unpaired group photos without references are included. MultiID-Bench establishes a standardized evaluation protocol for multi-identity generation. Beyond widely adopted metrics such as ID similarity (Schroff et al., 2015; Deng et al., 2019), it quantifies copy-paste artifacts by measuring distances between generated images, references, and ground truth. Evaluation on 12 state-of-the-art customization models highlights a clear trade-off between ID similarity and copy-paste artifacts (see Fig. 5).

Furthermore, we present **WithAnyone**, a novel identity customization model built on the FLUX (Batifol et al., 2025) architecture, as a step toward mitigating copy-paste artifacts. WithAnyone maintains state-of-the-art identity similarity (with regard to target image) while substantially reducing copy-paste, thereby breaking the long-observed trade-off between fidelity and artifacts. This advance is enabled by a paired-training strategy combined with an ID contrastive loss enhanced with a large negative pool, both made possible by our paired dataset. The labeled identities and their reference

images enable the construction of an extended negative pool (images of different identities), which provides stronger discrimination signals during optimization.

In summary, our main contributions are:

- **MultiID-2M:** A large-scale dataset of 500k group photos containing multiple identifiable celebrities, each with hundreds of reference images capturing diverse variations, along with 1.5M additional unpaired group photos. This resource supports pre-training and evaluation of multi-identity generation models.

- **MultiID-Bench:** A comprehensive benchmark with standardized evaluation protocols for identity customization, enabling systematic and intrinsic assessment of multi-identity image generation methods.

- **WithAnyone:** A novel ID customization model built on FLUX that achieves state-of-the-art performance, generating high-fidelity multi-identity images while mitigating copy-paste artifacts and enhancing visual quality.

## 2 RELATED WORK

**Single-ID Preservation.** Identity-preserving image generation is a core topic in customized synthesis (Wang et al., 2024a; Huang et al., 2024; Arar et al., 2024; Jones et al., 2024; Kumari et al., 2024; Zeng et al., 2023; Arar et al., 2023; Ma et al., 2024; Valevski et al., 2023; Wang et al., 2024b; Yan et al., 2023; Xiao et al., 2025; Wu et al., 2024b; Wang et al., 2024d; Chen et al., 2024; Hyung et al., 2024; Papantoniou et al., 2024). Many methods in the UNet/Stable Diffusion era inject learned embeddings (e.g., CLIP or ArcFace) via cross-attention or adapters (Ho et al., 2020; Ronneberger et al., 2015; Qian et al., 2024; Ye et al., 2023; Radford et al., 2021; Ren et al., 2023). With the rise of DiT-style backbones (Peebles & Xie, 2023; Esser et al., 2024; Labs, 2024) (e.g., SD3, FLUX), progress on ID preservation like PuLID (Guo et al., 2024), also attracts great attentions.

**Multi-ID Preservation.** Multi-ID preservation remains relatively underexplored. Some works target spatial control of multiple identities (Kim et al., 2024; He et al., 2024; Zhang et al., 2025), while others focus on identity fidelity. Methods such as XVerse (Chen et al., 2025) and UMO (Cheng et al., 2025) use VAE-derived face embeddings concatenated with model inputs, which can produce pixel-level copy-paste artifacts and reduce controllability. DynamicID (Hu et al., 2025a)[1] achieves improved controllability but is constrained by limited task-specific data and evaluation standards. Other general-purpose customization and editing models (Parmar et al., 2025; Mou et al., 2025; Patashnik et al., 2025; Wu et al., 2025d; Xiao et al., 2024; Wu et al., 2025b;c; Batifol et al., 2025; Wu et al., 2025a) can also synthesize images containing multiple identities, but their ID similarity are often compromised for generality.

**ID-Centric Datasets and Benchmarks.** Although numerous single-ID datasets (Karras et al., 2017; Wang et al., 2025) and multi-ID collections (Chu et al., 2024; Jiang et al., 2025b) exist, paired reference images are scarce, so reconstruction remains the dominant training objective for multi-ID datasets. Representative datasets are listed in Table 4. Evaluation protocols are underdeveloped: several works (e.g., PuLID (Guo et al., 2024), UniPortrait (He et al., 2024), and others (Xiao et al., 2025; Zhang et al., 2025)) construct test sets by sampling identities from CelebA (Liu et al., 2015), which undermines reproducibility. To address this, we release a curated multi-ID benchmark with standardized splits and comprehensive metrics to facilitate future research.

## 3 MULTIID-2M: PAIRED MULTI-PERSON DATASET CONSTRUCTION

MultiID-2M is a large-scale multi-person dataset constructed via a four-stage pipeline: (1) collect single-ID images from the web and construct a clean reference bank by clustering ArcFace (Deng et al., 2019) embeddings, yielding $\sim$1M reference images across $\sim$3k identities (averaging 400 per identity); (2) retrieve candidate group photos via multi-name and scene-aware queries and detect faces; (3) assign identities by matching ArcFace embeddings to single-ID cluster centers using cosine

---

[1]Excluded from our experiments due to unavailability of code and pretrained models.

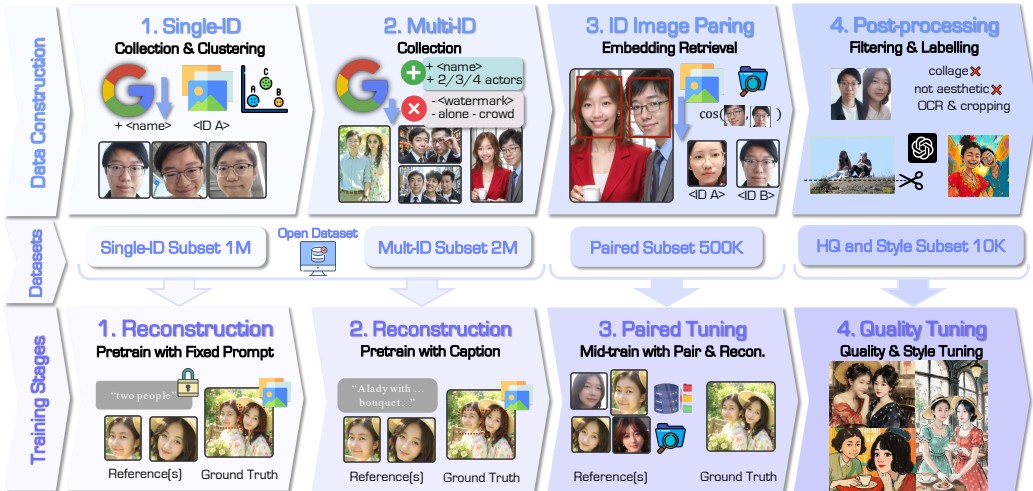

Figure 3: **Overview of WithAnyone.** It builds on a large-scale dataset, MultiID-2M, constructed through a four-step pipeline: (1) collect and cluster single-ID data based on identity similarity; (2) gather multi-ID data via targeted searches using desired identity names with negative keywords for filtering; (3) form image pairs by matching faces between single-ID and multi-ID data; and (4) apply post-processing for quality control and stylization. Training proceeds in four stages: (1) pre-train on single-ID, multi-ID, and open-domain images with fixed prompts; (2) train with image-caption supervision; (3) fine-tune with ID-paired data; and (4) perform quality tuning using a curated high-quality subset.

similarity (threshold 0.4); and (4) perform automated filtering and annotation, including Recognize Anything (Zhang et al., 2023b), aesthetic scoring (discus0434, 2023), OCR-based watermark/logo removal, and LLM-based caption generation (Bai et al., 2025). The final corpus comprises ~500k identified multi-ID images with matched references from the reference bank, as well as ~1.5M additional unidentified multi-ID images for reconstruction training, covering ~25k unique identities, with diverse nationalities and ethnicities.

Note that all images were collected from publicly accessible sources via search engines with explicit Creative Commons (CC) filters, restricted to publicly known figures, and limited to licenses permitting reuse and derivative works. We excluded content with restrictive or unclear terms and did not use private or login-restricted sources. In addition, no personal names or explicit identity labels were included in training, as individuals are represented only through anonymized internal identifiers. Further details of the data construction pipeline and dataset statistics are provided in Appendix C.

## 4 MULTIID-BENCH: COMPREHENSIVE ID CUSTOMIZATION EVALUATION

MultiID-Bench is a unified benchmark for group-photo (multi-ID) generation. It samples rare, long-tail identities with no overlap to training data, yielding 435 test cases. Each case consists of one ground-truth (GT) image containing 1–4 people, the corresponding 1–4 reference images as inputs, and a prompt describing the GT. Detailed statistics are provided in Appendix C.

Evaluation considers both identity fidelity and generation quality. Let $\mathbf{r}, \mathbf{t}, \mathbf{g}$ denote the face embeddings of the reference identity, the target (ground-truth), and the generated image, respectively. We define similarity between two embeddings as $\mathrm{Sim}(\mathbf{a}, \mathbf{b})$, specifically we term the generated image's face similarity with regard to GT as $\mathrm{Sim_{GT}}$, and to reference as $\mathrm{Sim_{Ref}}$,

$$\mathrm{Sim}(\mathbf{a}, \mathbf{b}) = \frac{\mathbf{a}^\top \mathbf{b}}{\|\mathbf{a}\| \, \|\mathbf{b}\|}, \quad \mathrm{Sim_{GT}} = \frac{\mathbf{g}^\top \mathbf{t}}{\|\mathbf{g}\| \, \|\mathbf{t}\|}, \quad \mathrm{Sim_{Ref}} = \frac{\mathbf{g}^\top \mathbf{r}}{\|\mathbf{g}\| \, \|\mathbf{r}\|}. \tag{1}$$

Prior works (Zhang et al., 2025; He et al., 2024; Guo et al., 2024; Cheng et al., 2025) has largely reported only $\mathrm{Sim_{Ref}}$, which inadvertently favors trivial copy-paste: directly replicating the reference appearance maximizes the score, even when the prompt specifies changes in pose, expression, or viewpoint. In contrast, MultiID-Bench uses $\mathrm{Sim_{GT}}$—the similarity to the ground-truth identity described by the prompt—as the primary metric. This design penalizes excessive copying when natural variations (e.g., pose, expression, occlusion) are expected, while rewarding faithful realization of the prompted scene.

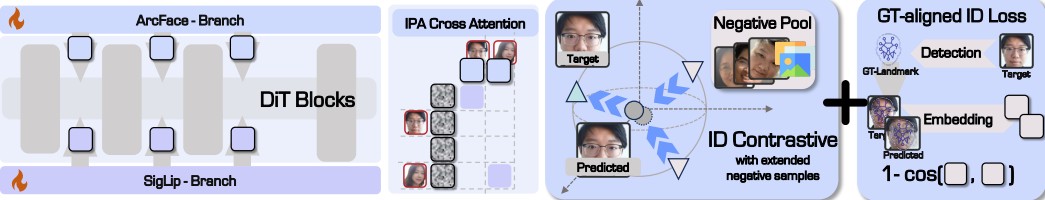

a **Model Architecture**      b **Training Objectives**

Figure 4: (a) **Architecture of WithAnyone**: Each reference is encoded by both a face-recognition network and a general image encoder, yielding identity-discriminative signals and complementary mid-level features. Face embeddings are restricted to attend only to image tokens within their corresponding face regions. (b) **Training Objectives of WithAnyone**: In addition to the diffusion loss, we incorporate an ID contrastive loss and a ground-truth–aligned ID loss, which together provide consistent and accurate identity supervision.

We define the angular distance as $\theta_{ab} = \arccos(\mathrm{Sim}(a, b))$ (geodesic distance on the unit sphere). The Copy-Paste metric is given by

$$M_{\mathrm{CP}}(\mathbf{g} \mid \mathbf{t}, \mathbf{r}) = \frac{\theta_{gt} - \theta_{gr}}{\max(\theta_{tr}, \varepsilon)} \in [-1, 1], \tag{2}$$

where $\varepsilon$ is a small constant for numerical stability. The metric thus captures the relative bias of $\mathbf{g}$ toward the reference $\mathbf{r}$ versus the ground truth $\mathbf{t}$, normalized by angular distance of $\mathbf{r}$ and $\mathbf{t}$. A score of 1 means $\mathbf{g}$ fully coincides with the reference (perfect copy-paste), while $-1$ means full agreement with the ground truth.

We additionally report identity blending, prompt fidelity (CLIP I/T), and aesthetics; formal definitions and further details are provided in Appendix D.

## 5   WITHANYONE: CONTROLLABLE AND ID-CONSISTENT GENERATION

Building on the scale and paired-reference supervision of the MultiID-2M, we devise training strategies and tailored objectives that transcend reconstruction to enable robust, identity-conditioned synthesis. This rich, identity-labeled supervision not only substantially improves identity fidelity but also suppresses trivial copy–paste artifacts and affords finer control over multi-identity composition. Motivated by these advantages, we introduce WithAnyone — a unified architecture and training recipe designed for controllable, high-fidelity multi-ID generation. Architectural schematics and implementation details are provided in Fig. 4 and Appendix E.

### 5.1   TRAINING OBJECTIVES

**Diffusion Loss.** We adopt the mini-batch empirical flow-matching loss. For each batch, we sample a data latent $x_1 \sim p_{\mathrm{data}}$, Gaussian noise $x_0 \sim \mathcal{N}(0, I)$, and a timestep $t \sim \mathcal{U}(0, 1)$. We then form the interpolated latent $x_t = (1 - t)x_0 + tx_1$ and regress the target velocity $(x_1 - x_0)$:

$$\mathcal{L}_{\mathrm{diff}} = \left\| v_\theta(x_t^{(i)}, t^{(i)}, c^{(i)}) - (x_1^{(i)} - x_0^{(i)}) \right\|_2^2, \tag{3}$$

where $c^{(i)}$ denotes the conditioning signal.

**Ground-truth-Aligned ID Loss.** Since ArcFace embedding requires landmark detection and alignment, directly extracting landmarks from $I_{\mathrm{gen}}$ is unreliable because generated images are obtained through noisy diffusion or one-step denoising. Prior methods compromise: PortraitBooth (Peng et al., 2024) applies the loss only at low noise levels ($t < 0.25$), discarding supervision at higher noise, while PuLID (Guo et al., 2024) fully denoises generated results at significant computational cost. In contrast, we align the generated image using GT landmarks, thereby avoiding noisy landmark extraction. We minimize the cosine distance between GT-aligned ArcFace embeddings of the generated and ground-truth (GT) faces:

$$\mathcal{L}_{\mathrm{ID}} = 1 - \cos(\mathbf{g}, \mathbf{t}) \tag{4}$$

where $\mathbf{g}$ and $\mathbf{t}$ are ArcFace embeddings of the generated and GT images. This design (1) enables applying the ID loss across all noise levels, (2) incurs negligible overhead throughout training, and (3) implicitly supervises generated landmarks. Ablation studies (Sec. 6.3) demonstrate more accurate

identity measurement and substantially improved identity preservation. Further explanation and notations are provided in Appendix E.1.

**ID Contrastive Loss With Extended Negatives.** To further strengthen identity preservation, we introduce an ID contrastive loss that explicitly pulls the generated image closer to its reference images in the face embedding space while pushing it away from other identities. The loss follows the InfoNCE (Oord et al., 2018) formulation:

$$\mathcal{L}_{\text{CL}} = -\log \frac{\exp(\cos(\mathbf{g}, \mathbf{t})/\tau)}{\sum_{j=1}^{M} \exp(\cos(\mathbf{g}, \mathbf{n}_j))/\tau)}, \tag{5}$$

where $\mathbf{t}$ is the embedding of the target, $\mathbf{n}_j$ are embeddings of $M$ negatives from different identities, and $\tau$ is a temperature hyperparameter. This formulation relies on ID-labeled datasets, which make it possible to draw thousands of negatives per sample from the reference bank, thereby greatly enriching the diversity of negative examples.

The overall training objective is a weighted sum of the above losses:

$$\mathcal{L} = \mathcal{L}_{\text{diff}} + \lambda_{\text{ID}}\mathcal{L}_{\text{ID}} + \lambda_{\text{CL}}\mathcal{L}_{\text{CL}}, \tag{6}$$

where $\lambda_{\text{ID}}$ and $\lambda_{\text{CL}}$ are hyper-parameters controlling the contributions of the ID loss and contrastive loss, respectively. Both are set to $0.1$ across all training phases described below.

## 5.2 TRAINING PIPELINE

Copy–paste artifacts largely arise from reconstruction-only training, which encourages models to replicate the reference image rather than learn robust identity-conditioned generation. Leveraging our paired dataset, we employ a four-phase training pipeline that gradually transitions the objective from reconstruction toward controllable, identity-preserving synthesis.

**Phase 1: Reconstruction pre-training with fixed prompt.** We begin with reconstruction pre-training to initialize the backbone, as this task is simpler than full identity-conditioned generation and can exploit large-scale unlabeled data. For the first few thousand steps, the caption is fixed to a constant dummy prompt (e.g., "two people"), ensuring the model prioritizes learning the identity-conditioning pathway rather than drifting toward text-conditioned styling. The full MultiID-2M is used in this phase, which typically lasts for 20k steps, at which point the model achieves satisfactory identity similarity. To further enhance data diversity, CelebA-HQ (Karras et al., 2017), FFHQ (Karras et al., 2019), and a subset of FaceID-6M (Wang et al., 2025) are also incorporated.

**Phase 2: Reconstruction pre-training with full captions.** This phase aligns identity learning with text-conditioned generation and lasts for an additional 40k steps, during which the model reaches peak identity similarity.

Table 1: **Quantitative comparison on the single-person subset of MultiID-Bench and OmniContext.** ▇, ▇, and ▇ indicate the first-, second-, and third-best performance, respectively. For Copy-Paste ranking, only cases with $\text{Sim}(\text{GT}) > 0.40$ are considered.

a **MultiID-Bench**

| Method | Identity Metrics | | | Generation Quality | | |
|---|---|---|---|---|---|---|
| | Sim(GT) ↑ | Sim(Ref) ↑ | CP ↓ | CLIP-I ↑ | CLIP-T ↑ | Aes ↑ |
| DreamO | 0.454 | 0.694 | 0.303 | 0.793 | 0.322 | 4.877 |
| OmniGen | 0.398 | 0.602 | 0.248 | 0.780 | 0.317 | 5.069 |
| OmniGen2 | 0.365 | 0.475 | 0.142 | 0.787 | 0.331 | 4.991 |
| FLUX.1 Kontext | 0.324 | 0.408 | 0.099 | 0.755 | 0.327 | 5.319 |
| Qwen-Image-Edit | 0.324 | 0.409 | 0.093 | 0.776 | 0.316 | 5.056 |
| GPT-4o Native | 0.425 | 0.579 | 0.178 | 0.794 | 0.311 | 5.344 |
| UNO | 0.304 | 0.428 | 0.141 | 0.765 | 0.314 | 4.923 |
| USO | 0.401 | 0.635 | 0.286 | 0.790 | 0.329 | 5.077 |
| UMO | 0.458 | 0.732 | 0.359 | 0.783 | 0.305 | 4.850 |
| UniPortrait | 0.447 | 0.677 | 0.265 | 0.793 | 0.319 | 5.018 |
| ID-Patch | 0.426 | 0.633 | 0.231 | 0.792 | 0.312 | 4.900 |
| InfU | 0.439 | 0.630 | 0.233 | 0.772 | 0.328 | 5.359 |
| PuLID | 0.452 | 0.705 | 0.315 | 0.779 | 0.305 | 4.839 |
| InstantID | 0.464 | 0.734 | 0.337 | 0.764 | 0.295 | 5.255 |
| Ours | 0.460 | 0.578 | 0.144 | 0.798 | 0.313 | 4.783 |
| GT | 1.000 | 0.521 | -0.999 | N/A | N/A | N/A |
| Ref | 0.521 | 1.000 | 0.999 | N/A | N/A | N/A |

b **OmniContext Single Character Subset**

| Method | Quality Metrics | | Overall |
|---|---|---|---|
| | PF ↑ | SC ↑ | Overall ↑ |
| DreamO | 8.13 | 7.09 | 7.02 |
| OmniGen | 7.50 | 5.52 | 5.47 |
| OmniGen2 | 8.64 | 8.50 | 8.34 |
| FLUX.1 Kontext | 7.72 | 8.60 | 7.94 |
| Qwen-Image-Edit | 7.66 | 8.16 | 7.51 |
| GPT-4o Native | 7.98 | 9.06 | 8.12 |
| UNO | 7.22 | 7.72 | 7.04 |
| USO | 6.96 | 7.88 | 6.70 |
| UMO | 6.56 | 7.92 | 6.79 |
| UniPortrait | 6.62 | 6.00 | 5.55 |
| ID-Patch | N/A | N/A | N/A |
| InfU | 7.69 | 4.62 | 4.70 |
| PuLID | 6.62 | 6.83 | 5.78 |
| InstantID | 4.89 | 5.49 | 4.35 |
| Ours | 7.43 | 7.04 | 6.52 |

**Phase 3: Paired tuning.** To suppress trivial copy–paste behavior, we replace $50\%$ of the training samples with paired instances drawn from the 500k labeled images in MultiID-2M. For each paired

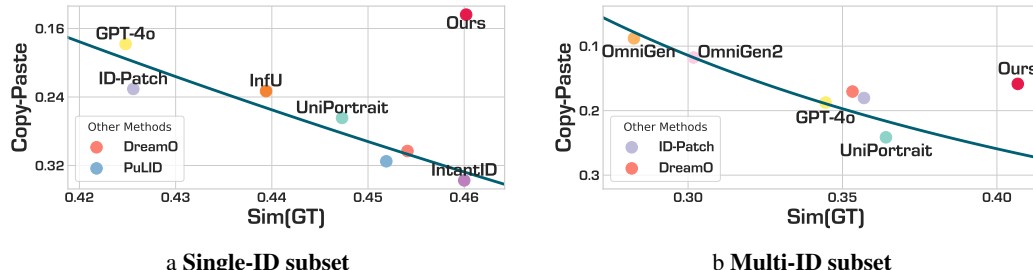

|  | a **Single-ID subset** |  | b **Multi-ID subset** |

Figure 5: **Face Similarity vs. Copy-Paste Effect (y-axis descending order).** Except for WithAnyone, all other models lie approximately on a fitted curve, reflecting a clear trade-off between face similarity and copy-paste. The upper-right corner represents the desired region.

sample, instead of using the same image as both input and target, we randomly select one reference image from the identity's reference set and another distinct image of the same identity as the target. This perturbation breaks the shortcut of direct duplication and compels the model to rely on high-level identity embeddings rather than low-level copying.

**Phase 4: Quality tuning.** Finally, we fine-tune on a curated high-quality subset augmented with generated stylized variants to (i) enhance perceptual fidelity and (ii) improve style robustness and transferability. This phase refines texture, lighting, and stylistic adaptability while preserving the strong identity consistency established in earlier phases.

# 6 EXPERIMENTS

In this section, we present a comprehensive evaluation of baselines and our WithAnyone model on the proposed MultiID-Bench.

**Baselines.** We evaluate two categories of baseline methods: general customization models and face customization methods. The general customization models include OmniGen (Xiao et al., 2024), OmniGen2 (Wu et al., 2025b), Qwen-Image-Edit (Wu et al., 2025a), FLUX.1 Kontext (Batifol et al., 2025), UNO (Wu et al., 2025d), USO (Wu et al., 2025c), UMO (Cheng et al., 2025), and native GPT-4o-Image (OpenAI, 2025). The face customization methods include UniPortrait (He et al., 2024), ID-Patch (Zhang et al., 2025), PuLID (Guo et al., 2024) (referring to its FLUX (Labs, 2024) implementation throughout this paper), and InstantID (Wang et al., 2024c). All models were evaluated on the single-person subset of the benchmark, while only those supporting multi-ID generation were additionally tested on the multi-person subset. Further implementation details are provided in Appendix F.1.

Table 2: **Quantitative comparison on the multi-person subset of MultiID-Bench.** ▇, ▇, and ▢ indicate the first-, second-, and third-best performance, respectively. For Copy-Paste ranking, only cases with $\text{Sim}(\text{GT}) > 0.35$ are considered. GPT exhibits prior knowledge of identities from TV series in subsets with more than two IDs, leading to abnormally high similarity scores.

a **2-people Subset**

| Method | Identity Metrics | | | | Generation Quality | | |
|---|---|---|---|---|---|---|---|
|  | Sim(GT)↑ | Sim(Ref)↑ | CP↓ | Bld↓ | CLIP-I↑ | CLIP-T↑ | Aes↑ |
| DreamO | 0.359 | 0.514 | 0.179 | 0.105 | 0.763 | 0.319 | 4.764 |
| OmniGen | 0.345 | 0.529 | 0.209 | 0.110 | 0.750 | 0.326 | 5.152 |
| OmniGen2 | 0.283 | 0.353 | 0.081 | 0.112 | 0.763 | 0.334 | 4.547 |
| GPT | 0.332 | 0.400 | 0.061 | 0.092 | 0.774 | 0.328 | 5.676 |
| UNO | 0.223 | 0.274 | 0.043 | 0.082 | 0.735 | 0.325 | 4.805 |
| UMO | 0.328 | 0.491 | 0.176 | 0.111 | 0.743 | 0.316 | 4.772 |
| UniPortrait | 0.367 | 0.601 | 0.254 | 0.075 | 0.750 | 0.323 | 5.187 |
| ID-Patch | 0.350 | 0.517 | 0.183 | 0.085 | 0.767 | 0.326 | 4.671 |
| Ours | 0.405 | 0.551 | 0.161 | 0.079 | 0.770 | 0.321 | 4.883 |

b **3-and-4-people Subset**

| Method | Identity Metrics | | | | Generation Quality | | |
|---|---|---|---|---|---|---|---|
|  | Sim(GT)↑ | Sim(Ref)↑ | CP↓ | Bld↓ | CLIP-I↑ | CLIP-T↑ | Aes↑ |
| DreamO | 0.311 | 0.427 | 0.116 | 0.081 | 0.709 | 0.317 | 4.695 |
| OmniGen | 0.345 | 0.529 | 0.209 | 0.110 | 0.750 | 0.326 | 5.152 |
| OmniGen2 | 0.288 | 0.374 | 0.099 | 0.071 | 0.734 | 0.329 | 4.664 |
| GPT | 0.445 | 0.484 | 0.048 | 0.044 | 0.815 | 0.320 | 5.647 |
| UNO | 0.228 | 0.276 | 0.046 | 0.065 | 0.717 | 0.319 | 4.880 |
| UMO | 0.318 | 0.465 | 0.180 | 0.070 | 0.717 | 0.309 | 4.946 |
| UniPortrait | 0.343 | 0.517 | 0.178 | 0.048 | 0.708 | 0.323 | 5.090 |
| ID-Patch | 0.379 | 0.543 | 0.195 | 0.059 | 0.781 | 0.329 | 4.547 |
| Ours | 0.414 | 0.561 | 0.171 | 0.045 | 0.771 | 0.325 | 4.955 |

## 6.1 QUANTITATIVE EVALUATION

The quantitative results are reported in Tables 1 and 2. We observe a clear trade-off between face similarity and copy-paste artifacts. As shown in Fig. 5, most methods align closely with a regression curve, where higher face similarity generally coincides with stronger copy-paste. This indicates that many existing models boost measured similarity by directly replicating reference facial features rather than synthesizing the identity. In contrast, WithAnyone deviates substantially from this curve,

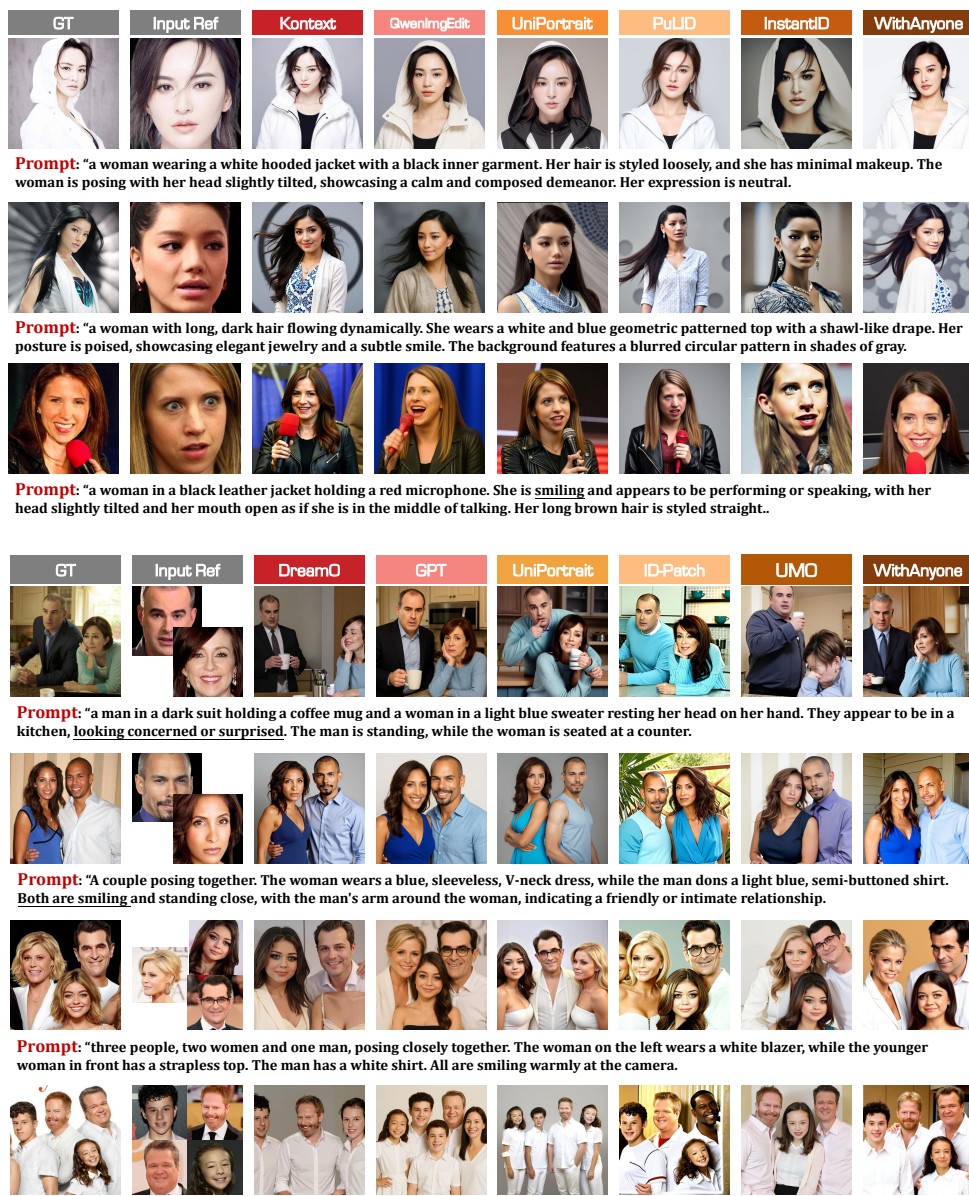

Figure 6: **Qualitative Results of Different Generation Methods.** The text prompt is extracted from the ground-truth image shown on the leftmost side.

achieving the highest face similarity with regard to GT while maintaining a markedly lower copy-paste score.

WithAnyone also achieves the highest score among ID-specific reference models on the OmniContext (Wu et al., 2025b) benchmark. However, VLMs (Bai et al., 2025; OpenAI, 2025) exhibit limited ability to distinguish individual identities and instead emphasize non-identity attributes such as pose, expression, or background. Despite that general customization and editing models often outperform face customization models on OmniContext, WithAnyone still has best performance among face customization models.

## 6.2 QUALITATIVE COMPARISON

To complement the quantitative results, Fig. 6 presents qualitative comparisons between our method, state-of-the-art general customization/editing models, and face customization generation models.

Table 3: **Ablation study.** ██, ██, and ░░ indicate the first-, second-, and third-best performance, respectively. We ablate paired data training phase (w/o Phase 3), the GT-aligned landmark ID loss (w/o GT-Align), and the use of extended negative samples in InfoNCE (w/o Ext. Neg.), and model trained on FFHQ only.

| | Ablation | Identity Metrics | | | Generation Quality | | |
|---|---|---|---|---|---|---|---|
| | | Sim(G) ↑ | Sim(R) ↑ | CP ↓ | CLIP-I ↑ | CLIP-T ↑ | Aes ↑ |
| Phases | w/o Phase 3 | 0.406 | 0.625 | 0.239 | 0.755 | 0.307 | 4.955 |
| Loss | w/o GT-Align | 0.385 | 0.549 | 0.175 | 0.763 | 0.317 | 4.754 |
| | w/o Ext. Neg. | 0.368 | 0.455 | 0.074 | 0.740 | 0.304 | 4.984 |
| Data | FFHQ only | 0.224 | 0.246 | 0.027 | 0.658 | 0.330 | 5.039 |
| Ours | Full Setting | 0.405 | 0.551 | 0.161 | 0.770 | 0.321 | 4.883 |

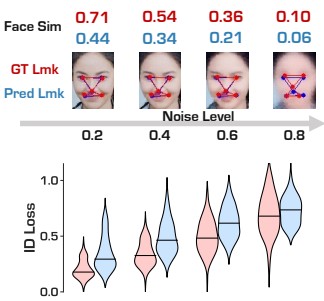

Figure 7: **Comparison of GT-aligned and prediction-aligned landmarks.**

It shows that identity consistency remains a significant weakness of general customization or editing models, consistent with our quantitative findings. Many VAE-based approaches—where references are encoded through a VAE, such as FLUX.1 Kontext and DreamO—tend to produce faces that either exhibit copy-paste artifacts or deviate markedly from the target identity. A likely reason is that VAE embeddings emphasize low-level features, leaving high-level semantic understanding to the diffusion backbone, which may not have been pre-trained for this task. ID-specific reference models also struggle with copy-paste artifacts. For example, they fail to make the subject smile when the reference image is neutral and often cannot adjust head pose or even eye gaze. In contrast, WithAnyone generates flexible, controllable faces while faithfully preserving identity.

## 6.3 ABLATION AND USER STUDIES

To better understand the contribution of each component in WithAnyone, we conduct ablation studies on the training strategy, the GT-aligned ID loss, the InfoNCE-based ID loss, and our dataset. Due to space constraints, we report the key results here, with additional analyses provided in Appendix G.

As shown in Table 3, the paired-data fine-tuning phase reduces copy-paste artifacts without diminishing similarity to the ground truth, while training on FFHQ performs significantly worse than on our curated dataset. Fig. 7 further demonstrates that the GT-aligned ID loss lowers denoising error at low noise levels and yields higher-variance, more informative gradients at high noise, thereby strengthening

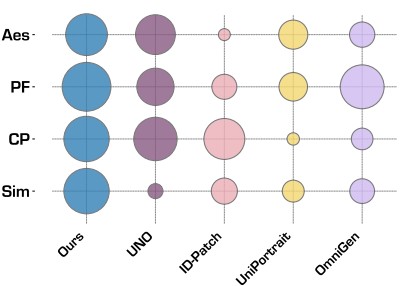

Figure 8: **User study.** Bigger bubbles indicate higher ranking and better performance.

identity learning. By ablating extended negatives, leaving only 63 negative samples from the batch (originally extended to 4096), the effectiveness of ID contrastive loss is greatly reduced. More ablation results can be found in Appendix G.

We conduct a user study to evaluate perceptual quality and identity preservation. Ten participants were recruited and asked to rank 230 groups of generated images according to four criteria: identity similarity, presence of copy-paste artifacts, prompt adherence, and aesthetics. The results, shown in Fig. 8, indicate that our method consistently achieves the highest average ranking across all dimensions, demonstrating both stronger identity preservation and superior visual quality. Moreover, the copy-paste metric exhibits a moderate positive correlation with human judgments, suggesting that it captures perceptually meaningful artifacts. Further details of the study design, ranking protocol, and statistical analysis are provided in Appendix H.

## 7 CONCLUSION

Copy-paste artifacts are a common limitation of identity customization methods, and face-similarity metrics often exacerbate the issue by implicitly rewarding direct copying. In this work, we identify and formally quantify this failure mode through MultiID-Bench, and propose targeted solutions. We curate MultiID-2M and develop training strategies and loss functions that explicitly discourage trivial

replication. Empirical evaluations demonstrate that WithAnyone significantly reduces copy-paste artifacts while maintaining—and in many cases improving—identity similarity, thereby breaking the long-standing trade-off between fidelity and copying. These results highlight a practical path toward more faithful, controllable, and robust identity customization.

## ETHICS STATEMENT AND DISCLAIMER

**Data Source**. The images used in this work were collected exclusively from publicly accessible sources through search engines that provide explicit filtering based on Creative Commons (CC) licensing. Our dataset focuses on publicly known figures, and we restricted data collection to images released under CC licenses that explicitly permit reuse and derivative works (e.g., CC-BY, CC-BY-SA, or CC0 where applicable). We excluded content with restrictive terms such as "NoAI", non-derivative, or unclear licensing conditions. No private datasets, login-restricted sources, or personally sensitive images were used. These license-based permissions provide authorization consistent with creator-defined reuse terms, and we follow applicable copyright and data-usage regulations to ensure responsible research practice.

**Anonymization**. To further mitigate privacy and misuse risks, we applied an anonymization procedure to the dataset processing and training pipeline. No personal names, textual identifiers, or explicit identity labels were included during training. All individuals are represented solely through internal numeric identifiers and corresponding ID embeddings, without any direct linkage to real-world names or metadata. The model therefore operates on abstract identity representations rather than explicit personal information, reducing the risk of unintended identity disclosure.

**Potential Ethical Risks and Mitigation**. Identity-consistent image generation is inherently dual-use. While enabling legitimate applications such as creative media and virtual avatars under proper authorization, it may also facilitate identity cloning, impersonation, misattribution, or deceptive synthetic media—particularly if applied without consent. To mitigate these risks, the models are released strictly for non-commercial academic research; training data is limited to publicly known figures under reuse-permitted licenses; and no personal names or explicit identity labels are used in training. We further recommend that downstream deployments implement consent verification, authorization controls, disclosure or watermarking mechanisms, and abuse monitoring. Responsible use must comply with applicable legal, institutional, and ethical standards.

**Disclaimer**. The *WithAnyone* models and associated datasets (the "Project") are provided solely for research and non-commercial use under the FLUX.1 [dev] Non-Commercial License v1.1.1. All base models and third-party components remain subject to their original licenses. Any underlying content derived from publicly available sources remains the property of its respective rights holders, and no ownership, endorsement, or additional rights are claimed or granted. The Project is provided "as is" without warranties of any kind, express or implied. Users are solely responsible for ensuring compliance with all applicable laws, regulations, and third-party rights. The providers of this Project shall not be liable for any claims, damages, or losses arising from its use. Under no circumstances shall the authors or the affiliated organization be liable for any claims, damages, losses, or other liabilities arising from or related to the use of the Dataset.

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

## A    FAMILY OF WITHANYONE

FLUX.1 comprises a family of models, including FLUX.1 Kontext (Batifol et al., 2025) and FLUX.1 Krea (Labs, 2025). Krea is a text-to-image model with improved real-person face generation, whereas Kontext is an image-editing model that excels at making targeted adjustments while preserving the rest of the image. However, as reported in Table 1, Kontext shows limited consistency with the reference face identity.

Our method, WithAnyone, can be seamlessly integrated into Kontext for the face customization downstream tasks like face swapping. As illustrated in Fig. 9, WithAnyone effectively injects identity information from the reference images into the target image.

The overall training pipeline follows the procedure described in Sec. 5, with a single modification: the input image provided to Kontext (whose tokens are concatenated with the noisy latent at each denoising step) is set to the target image with the face region blurred.

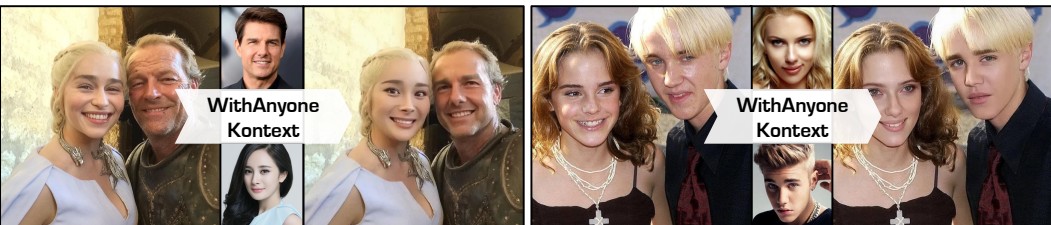

Figure 9: **Application of WithAnyone-Kontext.** Marrying editing models, WithAnyone is capable of face editing given customization references.

## B    GALLERIES OF WITHANYONE

We show more results of WithAnyone in Fig. 10, Fig. 11, and Fig. 12.

## C    MULTIID-2M CONSTRUCTION DETAILS

To fill in the void left by the lack of publicly available multi-ID datasets, a data constraction pipeline is proposed to create a large-scale dataset of multi-person images with paired identity references for identities on the data record. Based on this pipeline, 500k group photo images are collected, featuring 3k identities, each with hundreds of single-ID reference images. Another 1M images that cannot be identified are also included in the dataset for image reconstruction training purpose for image reconstruction training purpose.

### C.1    DATASET CONSTRUCTION PIPELINE

The pipeline contains four steps, as shown in Fig. 3. The detailed pipeline are as follows.

**Single-ID images.** To construct a ID reference set, single-ID images were collected from the web using celebrity names as search queries on Google Images. For each image, facial features were extracted with ArcFace (Ren et al., 2023), ensuring that only images containing exactly one face were retained. To remove outliers, DBSCAN (Schubert et al., 2017) clustering was applied to the embeddings for each celebrity, resulting in a set of cluster centers and hundreds of reference images per identity. This process established a reliable reference set for each unique identity. Human review confirms the accuracy of the ID bank built in this step.

**Multi-ID images**. To achieve best searching efficiency, group photos were obtained using more complex queries that combined multiple celebrity names, keywords indicating the number of people (e.g., "two celebrities"), scene descriptors (e.g., "award ceremony"), and negative keywords to filter out irrelevant results. ArcFace embeddings were extracted for these images, yielding a large pool of candidate multi-ID images. At this stage, the dataset comprised more than 20 million images.

**Retrieval.** To provide ID reference for the multi-ID images, it is necessary to retrieve the IDs on it. All single-ID cluster centers were aggregated into an embedding matrix. For each detected face

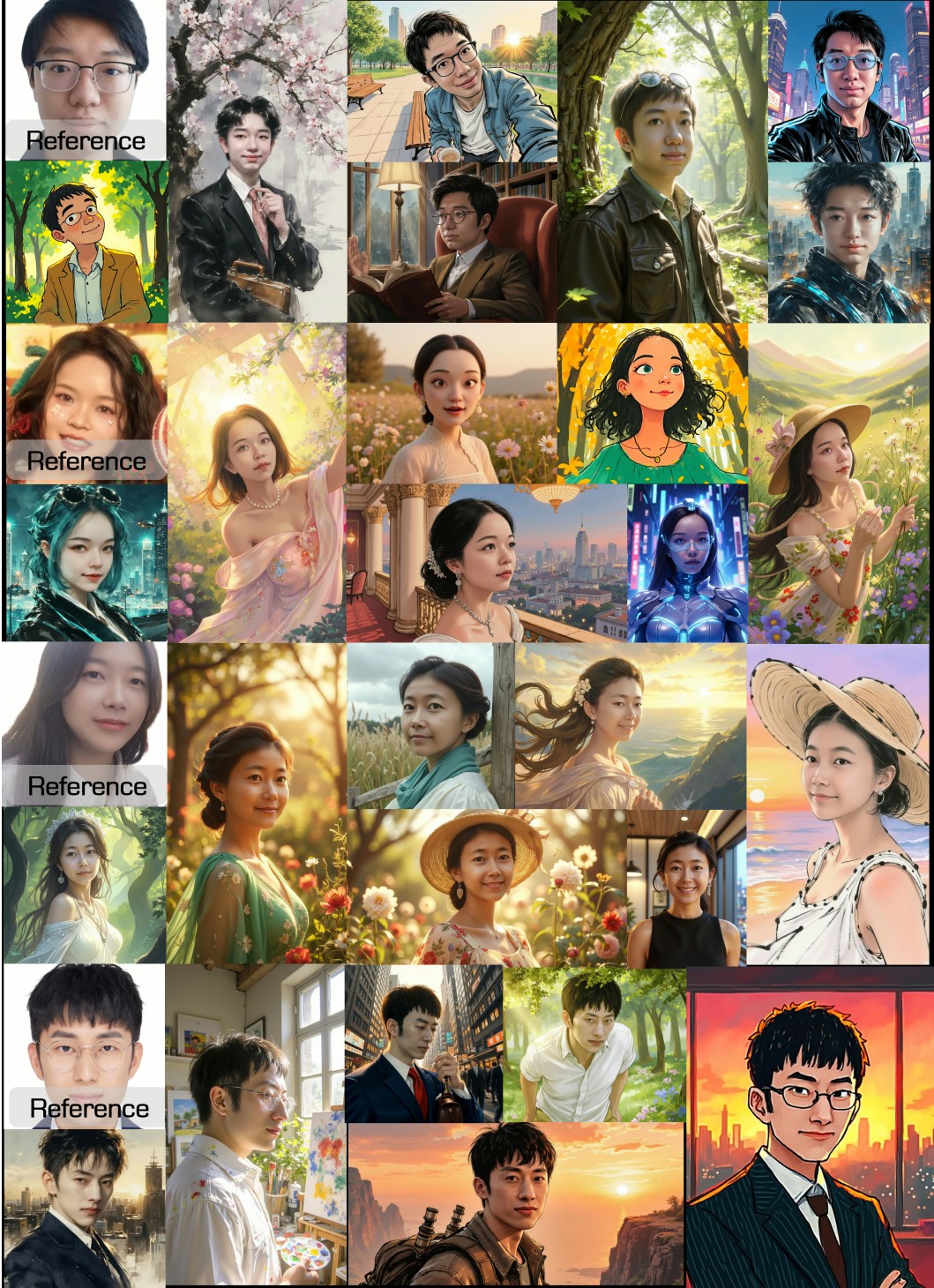

Figure 10: **Galleries of Single-ID Generation**.

in every multi-ID image, its ArcFace embedding was compared to all single-ID cluster centers to determine identity. The similarity between two embeddings was calculated as:

$$\text{sim}(id_1, id_2) = \cos(f(id_1), f(id_2)) \tag{7}$$

where $id_1$ and $id_2$ denote two faces, and $f$ is the ArcFace embedding network.

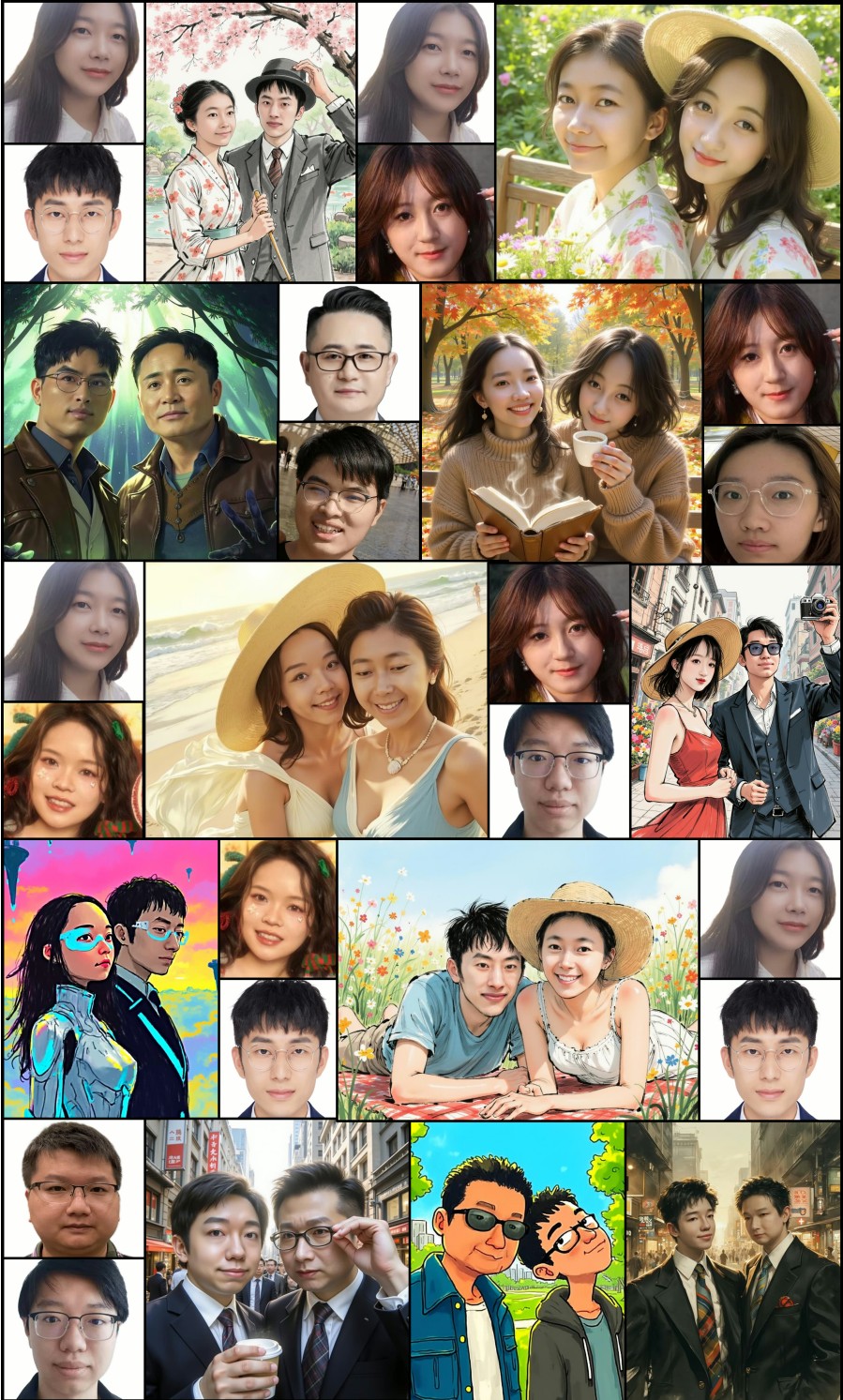

Figure 11: **Galleries of 2-person Generation**.

Each face in a multi-ID image was assigned the identity of the single-ID cluster center with the highest similarity, provided the similarity exceeded a predefined threshold (0.5). This approach enabled accurate and automated identity assignment in group images and facilitated retrieval of corresponding reference images.

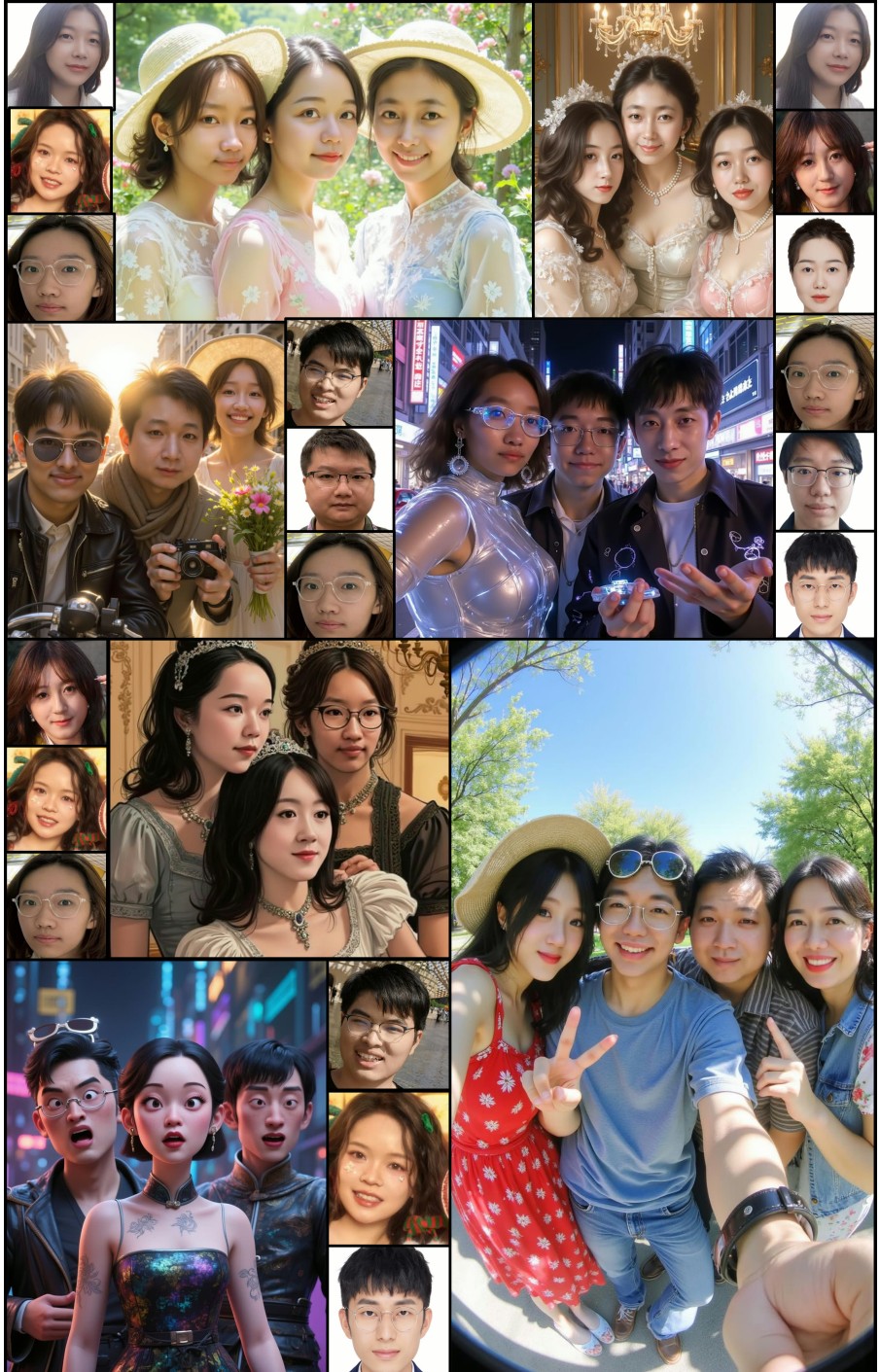

Figure 12: **Galleries of 3-to-4-person Generation**.

**Filtering and labelling.** To further improve dataset quality, a series of annotation and filtering steps were applied. The Recognize Anything model (Zhang et al., 2023b), an aesthetic score predictor (discus0434, 2023), and other auxiliary tools were used for annotation. Images with low aesthetic scores or those identified as collages rather than genuine group photos were excluded. Optical Character Recognition (OCR) tools detected watermarks and logos, which were cropped out when possible; otherwise, the images were discarded. Finally, descriptive captions were generated for the images using a large language model, enriching the dataset with textual information.

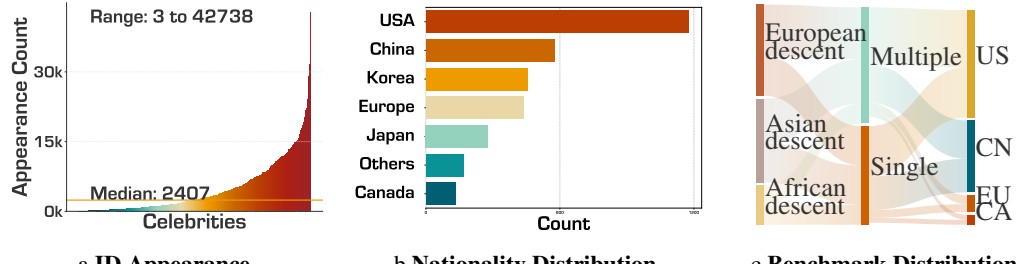

a **ID Appearance.**   b **Nationality Distribution.**   c **Benchmark Distribution.**

Figure 13: **Overview of Dataset Distributions.** (a) ID appearance distribution for the subset of one nation: the x-axis represents celebrities, sorted by the number of images in which they appear. (b) Nationality distribution: celebrities in our dataset come from over 10 countries, with most data sourced from China and the USA. (c) Word cloud of the most frequent words in the captions.

So far, a dataset with three parts is obtained: (1) 1M single-ID images as reference bank, or single-ID cross-paired training; (2) 500k paired multi-ID images with identified persons; (3) 1M unpaired multi-ID images, which can be used for training scenario without the need of references, such as reconstruction.

## C.2  DATASET STATISTICS

Comprehensive statistics of the dataset are provided in Fig. 14, including the distribution of nationalities, the count of appearances per identity, and a word cloud illustrating the most frequent terms in the generated image captions, offering insights into the diversity and richness of the dataset. A long-tail distribution is observed in the count of appearances per identity in Fig. 14a, with a few identities appearing frequently while many others are less common. This provide a diverse set of identities, as well as a perfect test dataset without identity interaction with the training set. Fig. 14b and Fig. 13c illustrate MultiID-2M's nationality distribution and action diversity respectively. The comparison between the proposed dataset and existing multi-ID datasets are listed in Table 4, highlighting MultiID-2M's outstanding volume and paired references.

Table 4: **Statistic Comparison for Identity-Centric Datasets**. **#Img** refers to total scale of the dataset; **#Paired** refers to paired group image number; **#Img / ID** indicates number of reference image for each single ID; **#ID / Img** means number of IDs appears on group photos.

| Dataset | #Img | #Paired | #Img / ID | #ID / Img |
|---|---|---|---|---|
| IMAGO (Stacchio et al., 2020) | 80k | 0 | 0 | - |
| MHP (Chu et al., 2024) | 5k | 0 | 0 | $2 - 10$ |
| PIPA (Zhang et al., 2015) | 40k | 40k | cross | $1 - 10$ |
| HumanRef (Jiang et al., 2025b) | 36k | 36 | 1+ | $1 - 14+$ |
| Celebrity Together (Zhong et al., 2018) | 194k | 0 | 0 | $1 - 5$ |
| PhotoMaker (Li et al., 2024) (closed source) | 1.3M | 1.3M | 100 | 1 |
| **MultiID-2M** | **2M** | **500k** | **100+** | $1 - 5$ |

## D  BENCHMARK AND METRICS DETAILS

Most existing methods are evaluated on privately curated test sets that are seldom released, and even when datasets are shared, the accompanying evaluation protocols vary widely. For example, ID-Patch (Zhang et al., 2025) and UniPortrait (He et al., 2024) measure identity similarity using ArcFace embeddings, whereas UNO (Wu et al., 2025d) relies on DINO (Oquab et al., 2023) and CLIP similarity scores. This heterogeneity—together with the common practice of reporting only the cosine similarity between matched ArcFace embeddings—fails to capture more nuanced insights and can even encourage degenerate behavior in which models produce images that are effectively "copy-pastes" of the reference photos.

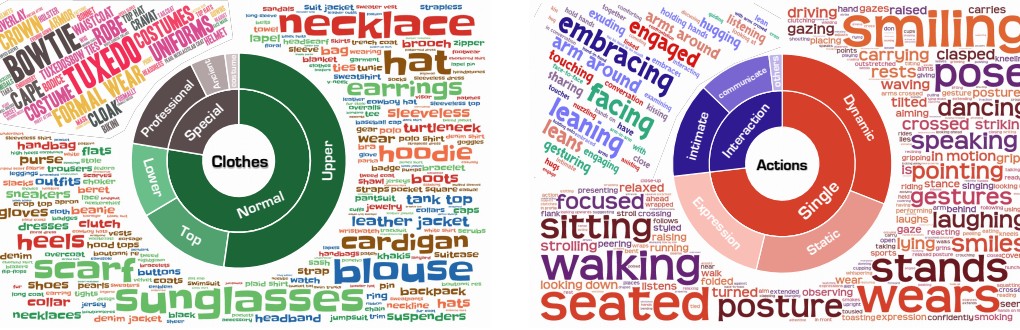

a **Clothes & Accessories Distribution.**        b **Action Distribution.**

Figure 14: **Distribution of Clothes and Action Labels of Proposed Dataset.**

In this work, MultiID-Bench is introduced as a unified and extensible evaluation framework for group photo (multi-ID) generation. It standardizes assessment along two complementary axes: (i) identity fidelity (preserving each target identity without unintended copying and blending), and (ii) generation quality (semantic faithfulness to the prompt/ground truth and overall aesthetic quality).

The data used in MultiID-Bench are drawn from the long-tail portion of MultiID-2M. We first select the least frequent identities and gather all images containing them. To prevent information leakage, the training split is filtered to ensure zero identity overlap with the benchmark set. The final benchmark contains 435 samples; each sample provides 1–4 reference identities (with their images), a corresponding ground-truth image, and a text prompt describing that ground-truth scene.

**Identity Blending.** In the similarity matrix, the off-diagonal elements correspond to the similarity between different identities. The average of the diagonal elements is used as the metric for identity fidelity, and the average of the off-diagonal elements serves as the metric for identity blending, as in Eq. 8.

$$\mathrm{M}_{\mathrm{Bld}}(x^g, x^t) = \frac{1}{N^2 - N} \sum_{i=1}^{N} \sum_{j=1, j \neq i}^{N} \cos(g_i, t_j) \qquad (8)$$

where $g_i$ is the embedding of the $i$-th face in the generated image $x^g$, and $t_j$ is the embedding of the $j$-th face in the ground-truth image $x^t$. A lower value indicates less unintended blending between different identities, which is desirable.

**Generation quality.** The overall generation quality is evaluated based on CLIP-I and CLIP-T, which are the de facto standards for evaluating the prompt-following capability (Radford et al., 2021), are employed to measure the cosine similarity in the CLIP embedding space between the generated image and the ground truth image or caption. Additionally, an aesthetic score model (discus0434, 2023) is used to assess the aesthetic quality of the generated images.

## E  MODEL FRAMEWORK DETAILS

We follow prior work (Ye et al., 2023; Guo et al., 2024) and integrate a lightweight identity adapter into the diffusion backbone. Identity embeddings are injected by cross-attention so that the base generative prior is preserved while controllable identity signals are added.

**Face embedding.** Each reference face is first encoded by ArcFace, producing a $1 \times 512$ identity embedding. To match the tokenized latent space of the DiT backbone, this vector is projected with a multi-layer perceptron (MLP) into 8 tokens of dimension 3072 (i.e., an $8 \times 3072$ tensor). This tokenization provides sufficient capacity for the cross-attention layers to integrate identity cues without overwhelming the generative context.

**Controllable attribute retention.** Completely suppressing copy-like behavior is not always desirable: users sometimes expect certain mid-level appearance attributes (e.g., hairstyle, accessories) to be preserved. ArcFace focuses on high-level, identity-discriminative geometry and texture cues but

omits many mid-level semantic factors. To expose controllable retention of such attributes when needed, we optionally incorporate SigLIP (Zhai et al., 2023) as a secondary encoder. SigLIP provides more semantically entangled representations, enabling selective transfer of style-relevant traits while ArcFace anchors identity fidelity.

**Attention mask and location control.** To further improve identity disentanglement and precise localization in the generated images, an attention mask and location control mechanism are incorporated. Specifically, ground-truth facial bounding boxes are extracted from the training data and used to generate binary attention masks. These masks are applied to the attention layers of the backbone model, ensuring that each reference token only attends to its corresponding face region in the image, providing location control at the same time.

**Feature injection.** After each transformer block of the DiT backbone, we inject face features through a cross-attention modulation:

$$H' = H + \lambda_{\text{id}} \, \text{softmax}\left(\frac{(HW_Q)(EW_K)^\top}{\sqrt{d}} + M\right)(EW_V), \tag{9}$$

where $H$ denotes the current hidden tokens, $E$ the stacked face-embedding tokens, and $W_Q, W_K, W_V$ the projection matrices; $d$ is the query/key dimension, and $\lambda_{\text{id}} = 1.0$ during training. When SigLIP is enabled, its tokens are processed by a parallel cross-attention with an independent scaling coefficient.

### E.1 DETAILS OF THE GT-ALIGNED ID LOSS

To compute the ID Loss, the predicted velocity must first be converted into a fully denoised latent representation, which is then decoded into the pixel space. Typically, this process requires around 20 iterative denoising steps, leading to an unacceptable computational overhead. PortraitBooth (Peng et al., 2024) mitigates this issue by applying the loss only at low noise levels ($t < 0.25$), effectively discarding supervision at higher noise levels. In contrast, PuLID (Guo et al., 2024) performs full denoising with four iterative steps, incurring significant computational cost.

Since WithAnyone is based on FLUX and employs a flow-matching diffusion objective, where the model directly predicts the velocity field, the denoised latent can be obtained in a single step. Let $x_0$ denote pure noise and $x_1$ the clean latent. The noisy latent input at timestep $t$ during training is given by:

$$x_t = (1 - t)x_1 + tx_0, \tag{10}$$

where $t$ is the sampled timestep. The model predicts the velocity $\hat{v}$ corresponding to $x_0 - x_1$. Therefore, the predicted denoised latent can be expressed as:

$$\hat{x} = x_t - t\hat{v}. \tag{11}$$

Decoding $\hat{x}$ into pixel space yields the reconstructed image used for identity (ID) feature extraction in the face recognition network.

We find this implementation both highly efficient and effective. The visual quality of the predicted images during training is illustrated in Fig. 7. Since single-step denoising introduces minimal computational overhead, the ID Loss can be applied throughout the entire training process, rather than being restricted to the fine-tuning stage as in PuLID (Guo et al., 2024), or only in low timestep as in PortraitBooth (Peng et al., 2024).

Denoting the face recognition model as $f(\cdot, \cdot)$ (Arcface (Deng et al., 2019), in our case), and the coupled detection model as $g(\cdot)$ (RetinaFace (Deng et al., 2020)), the generated image as $\mathbf{G}$, and the ground-truth image as $\mathbf{T}$, a embedding extraction should be performed as follows:

$$\mathbf{g} = f(g(\mathbf{G}), \mathbf{G}), \quad \mathbf{t} = f(g(\mathbf{T}), \mathbf{T}), \tag{12}$$

where $g(\mathbf{G})$ and $g(\mathbf{T})$ are the detected landmarks, and $f(\cdot, \cdot)$ extracts the aligned face embedding. The ID loss in Guo et al. (2024) and is computed as:

$$\mathcal{L}_{\text{id}} = 1 - \cos(f(g(\mathbf{G}), \mathbf{G}), \mathbf{t} = f(g(\mathbf{T}), \mathbf{T})). \tag{13}$$

And our GT-aligned ID loss is computed as:

$$\mathcal{L}_{\text{id}} = 1 - \cos(f(g(\mathbf{T}), \mathbf{G}), \mathbf{t} = f(g(\mathbf{T}), \mathbf{T})). \tag{14}$$

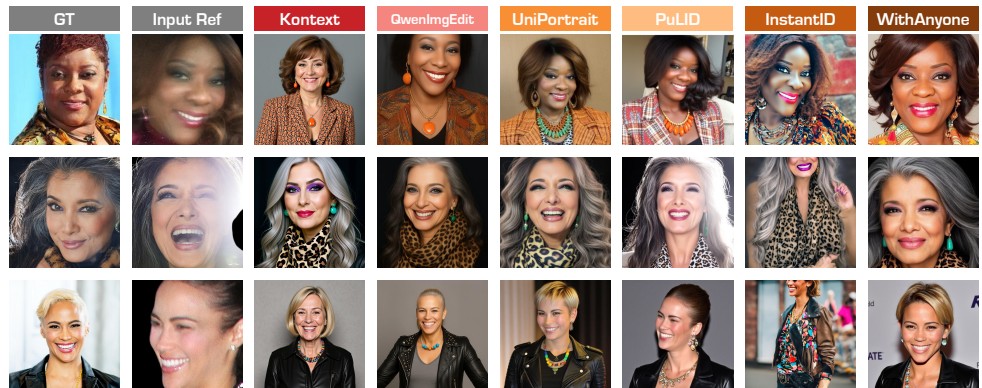

Figure 15: **Qualitative Results of Low Quality References.** WithAnyone is more robust when presented low-quality references like low-resolution, motion blur, or len-flare.

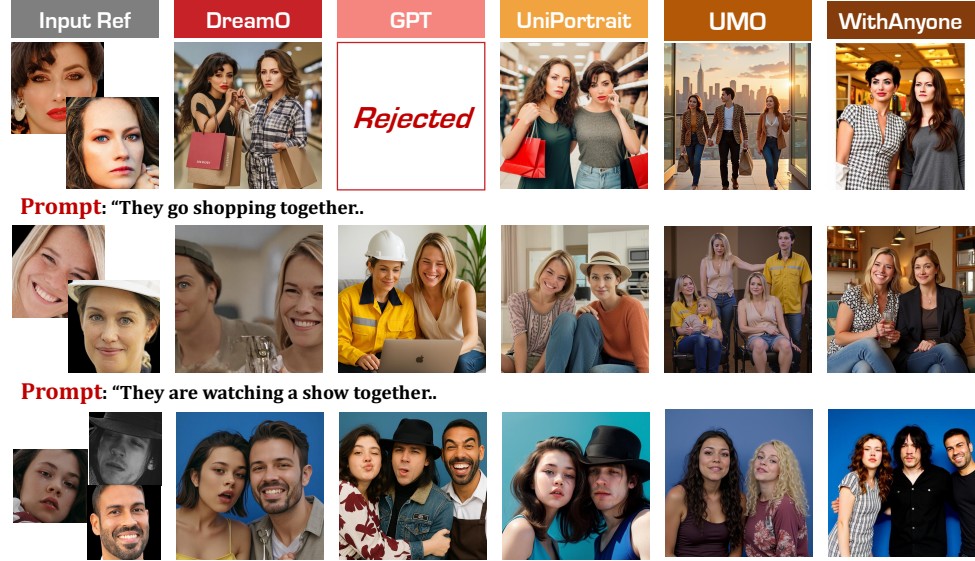

Figure 16: **Qualitative Results on Non-Celebrity-References.** WithAnyone is robust enough to generalize to non-celebrity identities. Cases are from OmniConetxt

## F   EXPERIMENTAL DETAILS

### F.1   IMPLEMENTATION DETAILS

WithAnyone is trained on 8 NVIDIA H100 GPUs, with a batch size of $4$ on each GPU. The learning rate is set to $1e^{-4}$, and the AdamW optimizer is employed with a weight decay of $0.01$. The pre-training phase runs for $60k$ steps, with a fixed prompt used during the first $20k$ steps. The subsequent paired-tuning phase lasts $30k$ steps: $50\%$ of the samples use paired (reference, ground-truth) data, while the remaining $50\%$ continue reconstruction training. Finally, a quality/style tuning stage of $10k$ steps is performed with a reduced learning rate of $1 \times 10^{-5}$.

For the extended ID contrastive loss, the target is used as the positve sample, while other IDs from samples in the same batch serve as negative samples. With the global batch size of 32, this yields less than a hundred negative samples. Extended negative samples are drawn from reference bank. If this ID is identified as one of the 3k ID in the reference bank, we simply omit its own ID and draw the from other IDs. If this ID is not identified, then it makes things easier – all the IDs in the reference bank can be used as negative samples.

For other baseline methods, official implementations and checkpoints (or API) are used with default settings. Methods are tested on MultiID-Bench and real-human subset of OmniContext (Wu et al.,

2025b). OmniContext uses Vision-Language Models (VLMs) to evaluate the prompt-following (PF) and subject-consistency (SC) of generated images. For reproducibility, the VLM is fixed to Qwen2.5-VL (Bai et al., 2025). ID-Patch (Zhang et al., 2025) requires pose condition, and we use the ground-truth pose for it.

Single face embedding model may induce biased evaluation on ID similarity, thus we average three de-facto face recognition models' consine similarity to compute the overall ID similarity metric, namely ArcFace (Deng et al., 2019), FaceNet (Schroff et al., 2015), and AdaFace (Kim et al., 2022).

## F.2  More Qualitative Results

**Results on Low Quality References.** We show more results on low-quality reference images in Fig. 15. While many ID-preserving methods even copy the low-quality artifacts like blurriness and flare into the generated images, WithAnyone successfully preserves the identity while generating high-quality images.

**Results on Non-Celebrity References.** We show more results on non-celebrity reference images in Fig. 16. WithAnyone shows consistent identity-preserving ability on non-celebrity references, demonstrating its generalization ability beyond celebrities commonly found in training data.

## F.3  More Discussion on the Quantitative Results

The performance of GPT on our 3-and-4-people subset offers a useful validation of our copy-paste metric, as shown in Table 2. This subset largely comprises group photographs from TV series that GPT may have encountered during pretraining, so GPT attains unusually high identity-similarity scores both to the ground truth (GT) and to the reference images. Actually, in one case GPT even generates an ID from the TV series that is not present in the reference images. This behaviour approximates an idealized scenario in which a model fully understands and faithfully reproduces the target identity: similarity to GT and to references are both high, and the copy-paste measure—the difference between distances to GT and to references—approaches zero. These observations are consistent with our metric design and support its ability to distinguish true identity understanding from trivial copy-and-paste replication.

We report the experimental limit in Table 1. If one model completely copy the reference image, $\text{Sim}_{\text{GT}} = 0.521$, $\text{Sim}_{\text{Ref}} = 1.0$, and copy-paste is 0.999, which aligns with the theoretical limit 1.0 of copy-paste.

The prompt-following ability is measured by CLIP-I and CLIP-T in our benchmark, and is judged by VLM in OmniContext. WithAnyonegains state-of-the-art performance in both metrics, and is ranked the highest in our user study. However, the credibility of CLIP scores and the aesthetic scores may be debated, as they are not always consistent with human perception.

## F.4  More Quantitative Results

Table 5: **Quantitative comparison on MultiID-Bench with different face recognition models**. ▨ ▨ ▨ indicate the first, second, third performance respectively. For Copy-Paste ranking, only cases with Sim(GT) higher than a threshold are compared. Thresholds for ArcFace, AdaFace and FaceNet are 0.35, 0.30, 0.45 respectively.

| Method | ArcFace | | | AdaFace | | | FaceNet | | |
|---|---|---|---|---|---|---|---|---|---|
| | Sim(GT) ↑ | Sim(Ref) ↑ | CP ↓ | Sim(GT) ↑ | Sim(Ref) ↑ | CP ↓ | Sim(GT) ↑ | Sim(Ref) ↑ | CP ↓ |
| DreamO | 0.315 | 0.476 | 0.197 | 0.286 | 0.455 | 0.177 | 0.480 | 0.607 | 0.151 |
| OmniGen | 0.296 | 0.453 | 0.186 | 0.277 | 0.462 | 0.187 | 0.464 | 0.673 | 0.250 |
| OmniGen2 | 0.193 | 0.269 | 0.089 | 0.197 | 0.275 | 0.078 | 0.461 | 0.516 | 0.070 |
| GPT | 0.261 | 0.314 | 0.059 | 0.245 | 0.310 | 0.059 | 0.491 | 0.576 | 0.092 |
| UNO | 0.130 | 0.174 | 0.045 | 0.151 | 0.194 | 0.039 | 0.408 | 0.469 | 0.056 |
| UMO | 0.278 | 0.420 | 0.168 | 0.266 | 0.439 | 0.173 | 0.448 | 0.610 | 0.188 |
| UniPortrait | 0.356 | 0.542 | 0.224 | 0.307 | 0.525 | 0.217 | 0.440 | 0.730 | 0.330 |
| ID-Patch | 0.302 | 0.440 | 0.162 | 0.284 | 0.459 | 0.175 | 0.470 | 0.644 | 0.209 |
| Ours | 0.388 | 0.544 | 0.195 | 0.338 | 0.485 | 0.150 | 0.472 | 0.568 | 0.095 |

Three face recognition models are used to compute the ID similarity metric in the main paper. We report the individual scores of each model in Table 5. The results are consistent across ArcFace Deng

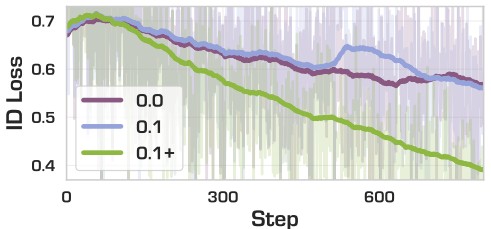
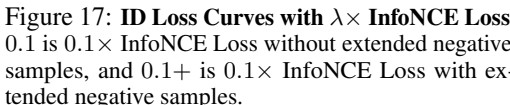
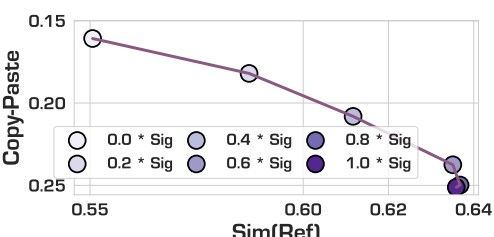

Figure 17: **ID Loss Curves with** $\lambda\times$ **InfoNCE Loss.** 0.1 is $0.1\times$ InfoNCE Loss without extended negative samples, and $0.1+$ is $0.1\times$ InfoNCE Loss with extended negative samples.

Figure 18: **Trade-off Curves** with $\lambda\times$ Siglip and $(1-\lambda)\times$ ArcFace signal.

et al. (2019) and AdaFace Kim et al. (2022), while FaceNet Schroff et al. (2015) shows slightly more similar scores among different methods. Overall, WithAnyone achieves SOTA performance in SOTA face recognition models.

## G    ABLATION STUDY DETAILS

Table 6: **Model Performance After Each Training Phases.** Scores of model checkpoints after each training phases are shown in this table. For stage 1, we present results both with fixed prompt (like "two people", "a portrait"), and without fixed prompt (original caption of the training record).

| Phase | Ablation | Identity Metrics | | | Generation Quality | | |
|---|---|---|---|---|---|---|---|
| | | Sim(G) ↑ | Sim(R) ↑ | CP ↓ | CLIP-I ↑ | CLIP-T ↑ | Aes ↑ |
| Phase 1 | w/o fixed txt | 0.277 | 0.350 | 0.066 | 0.727 | 0.337 | 4.715 |
| | w fixed txt | 0.305 | 0.428 | 0.104 | 0.758 | 0.315 | 5.082 |
| Phase 2 | - | 0.406 | 0.625 | 0.239 | 0.755 | 0.307 | 4.955 |
| Phase 3 | - | 0.409 | 0.572 | 0.168 | 0.765 | 0.324 | 4.854 |
| Phase 4 | - | 0.405 | 0.551 | 0.161 | 0.770 | 0.321 | 4.883 |

In this section, we systematically evaluate the impact of training strategy, GT-aligned ID-Loss, InfoNCE ID Loss, and our dataset construction. User study is also conducted to validate the consistency of the proposed metrics with human perception, as well as evaluate the human preference on different methods.

**SigLIP signal.** SigLIP (Zhai et al., 2023) signal is introduced to retain copy-paste effect when user tend to retain the features from reference images like hairstyle, accessories, etc. As shown in Fig. 18, increasing the SigLIP signal weight effectively amplifies the copy-paste effect while simultaneously boosting ID similarity to the reference images—exactly as expected, since stronger SigLIP guidance enforces tighter semantic alignment and transfers more fine-grained appearance cues (e.g., hairstyle, accessories, local textures).

**Training strategy.** Results after each training phases are shown in Table 6. To our insight, 1) phase 1 gives the model the basic ability of ID-preserving generation, and by fixing the prompt this process is accelerated. 2)Phase 2 further improves the ID consistency and reaches the highest similarity and copy-paste scores. 3) Phase 3 significantly reduces the copy-paste issue while maintaining (even sightly improving) the Sim_GT. 4) Phase 4 maintains the quantitative scores while making the model capable of generating style-varied images.

Fig. 19 shows the effect of phase 4 quality/style tuning. This phase effectively enables the model to generate images with diverse styles while preserving identity consistency.

**Dataset construction.** To validate the effectiveness of our dataset, we trained a model on FFHQ (Karras et al., 2019) using reconstruction training for the same number of steps. As shown in Table 3, the FFHQ-trained model performs poorly across all metrics. This likely stems from FFHQ's limited diversity and size, as it contains only 70k face-only portrait images.

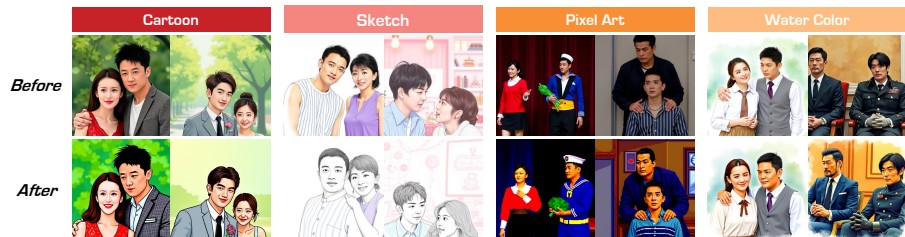

Figure 19: **Qualitative Results Before and After Stylized Quality Tuning (Phase 4).** Training phase 4 greatly improves the capability of generating stylized results.

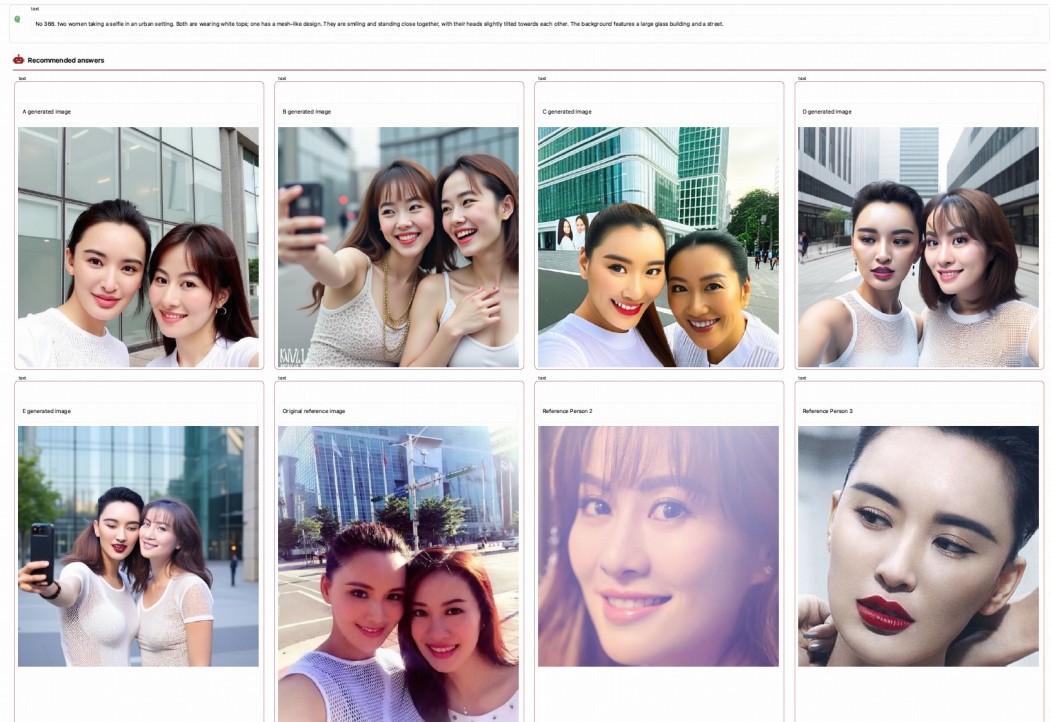

Figure 20: **User Study Interface**.

**GT-aligned ID-Loss.** We validate the GT-aligned ID-Loss with a simple experiment that visualizes predicted faces at different denoising time steps during training. As shown in Fig. 7, at low noise levels the GT-aligned ID-Loss is substantially lower than the loss computed using prediction-aligned landmarks, indicating that aligning faces to ground-truth landmarks reduces denoising error and yields a more accurate identity assessment. At high noise levels the GT-aligned ID-Loss shows greater variance, producing stronger and more informative gradients that help the model learn identity features.

**InfoNCE Loss.** The InfoNCE loss with extended negative samples is crucial for the convergence in the early training stage. We conduct a toy experiment with 1000 training samples, and record ID Loss curves with no InfoNCE loss, $0.1\times$ InfoNCE loss without extended negatives, and $0.1\times$ InfoNCE loss with extended negatives. As shown in Fig. 17, ID loss fits a lot faster with InfoNCE loss with extended negatives, demonstrating its effectiveness in accelerating training convergence. It also largely increases the ID similarity score, as shown in Table 3.

## H  USER STUDY DETAILS

Our user study is conducted with the same data samples and generated results in our quantitative experiments. Due to a tight financial budget, we randomly select 100 samples from single-person subset, 100 samples from 2-people subset, and all samples from 3-and-4 people subset. 10 participants

are recruited for the study, all of whom are trained with a brief tutorial to understand the task and evaluation criteria.

We illustrate the interface used in our user study in Fig. 20.

Table 7: **Correlation Statistics Between Machine Ranking and Human Ranking.** Reported values include Pearson's $r$, Spearman's $\rho$, and Kendall's $\tau$ with corresponding $p$-values.

| Dimension (N) | Pearson $r$ (p) | Spearman $\rho$ (p) | Kendall $\tau$ (p) |
|---|---|---|---|
| Copy-Paste | 0.4417 ($7.98e{-}48$) | 0.4535 ($1.26e{-}50$) | 0.3405 ($1.10e{-}46$) |
| ID Sim | 0.3254 ($1.54e{-}26$) | 0.3237 ($2.91e{-}26$) | 0.2423 ($1.11e{-}25$) |

## H.1 CORRELATION ANALYSIS

We analyze the correlation between our proposed metrics and user study results. As shown in Table 7, our copy-paste metric shows a moderate positive correlation with user ratings on copy-paste effect.

## H.2 PARTICIPANT INSTRUCTIONS

We provide the instructions for training the participants in the following table.

# I PROMPTS FOR LANGUAGE MODELS

Large language models (LLMs) and vision-language models (VLMs) are used in various stages of our work, including dataset captioning and OmniContext evaluation.

## I.1 DATASET CAPTIONING

Besides the system prompt, we design 6 different prompts to generate diverse captions for each image. 1 prompt is randomly selected for each image during captioning.

**Participant Instructions and Evaluation Procedure**

**Data source and task overview.**
Five different methods generated images under the following conditions:

- A single prompt that describes the "ground truth image."
- Between 1 and 4 people in the scene (most examples contain 1–2 people).

For each trial you will be shown the ground truth image, input images, and a generation instruction. Then you will observe five generated group-photo results (one per method) and rank them according to several evaluation dimensions. Use a 5-star scale where 5 stars = best and 1 star = worst. Please read the input image(s) and the editing instruction carefully before inspecting the generated results.

**Evaluation procedure (per-image ranking).**
Rank each generated image individually on the following criteria.

**Identity similarity**

- How well do the person(s) in the generated image resemble the person(s) in the ground truth image?
- Rank images by their resemblance to the ground truth image: the more the generated person(s) look like the original reference, the higher the rating.
- **Important:** When judging identity similarity, ignore factors such as image quality, rendering artifacts, or general aesthetics. Focus only on how much the person(s) resemble the original reference(s). Also, try to assess resemblance to the ground truth image as a whole, rather than comparing to any single separate "reference person n."

**Copy-and-paste effect (excessive mimicry of the reference)**

- Generated images should resemble the original reference but should not be direct copies of an individual reference photo.
- Evaluate whether the generated person appears to be directly copied from one of the reference images. Consider changes (or lack thereof) in **expression, head pose and orientation, facial expression/demeanor, and lighting/shading**.
- The lower the degree of direct copying (i.e., the less it looks like a pasted replica), the better. Rank according to the amount of change observed in the person(s): more natural variation (less copy-paste) should be ranked higher.

**Prompt following**

- Does the generated image reflect the content and constraints specified by the prompt/instruction?
- Rank images by prompt fidelity: the more faithfully the image follows the prompt, the higher the ranking.

**Aesthetics**

- Judge the overall visual quality and pleasantness of the generated image (e.g., smoothness of rendering, harmonious body poses and composition).
- Rank images by aesthetic quality: higher perceived visual quality receives higher ratings.

**Full Prompts for Dataset Captioning (6 variants)**

System Prompt: You are an advanced vision-language model tasked with generating accurate and comprehensive captions for images.

**Prompt 1:** Please provide a brief description of the image based on these guidelines:

1. Describe the clothing, accessories, or jewelry worn by the people in detail.
2. Describe the genders, actions, and posture of the individual in detail, focusing on what they are doing.
3. The description should be concise, with a maximum of 77 words.
4. Start with 'This image shows'

**Prompt 2:** Offer a short description of the image according to these rules:

1. Focus on details about clothing, accessories, or jewelry.
2. Focus on the gender, activity, and pose, and explain what the people is doing.
3. Keep the description within 77 words.
4. Begin the description with 'This image shows'

**Prompt 3:** Please describe the image briefly, following these instructions:

1. Provide a detailed description of the clothing or jewelry the person may be wearing.
2. Provide a detailed description of the two persons' gender, actions, and body position.
3. Limit the description to no more than 77 words.
4. Begin your description with 'This image shows'

**Prompt 4:** Describe the picture briefly according to these rules:

1. Provide a detailed description of the clothing, jewelry, or accessories of the individuals.
2. Focus on the two persons' gender, what they are doing, and their posture.
3. Keep the description concise, within a limit of 77 words.
4. Start your description with 'This image shows'

**Prompt 5:** Provide a short and precise description of the image based on the following guidelines:

1. Describe what the person is wearing or any accessories.
2. Focus on the gender, activities, and body posture of the person.
3. Ensure the description is no longer than 77 words.
4. Begin with 'This image shows'

**Prompt 6:** Briefly describe the image according to these instructions:

1. Provide a precise description of the clothing, jewelry, or other adornments of the people.
2. Focus on the person's gender, what they are doing, and their posture.
3. The description should not exceed 77 words.
4. Start with the phrase 'This image shows'

---

**Modified Prompt for OmniContext Evaluation (Face Identity Focus)**

Rate from 0 to 10:

**Task:** Evaluate how well the facial features in the final image match those of the individuals in the original reference images, as described in the instruction. Focus strictly on facial identity similarity; ignore hairstyle, clothing, body shape, background, and pose.

**Scoring Criteria**
- **0:** The facial features are completely different from those in the reference images.
- **1–3:** The facial features have minimal similarity with only one or two matching elements.
- **4–6:** The facial features have moderate similarity but several important differences remain.
- **7–9:** The facial features are highly similar with only minor discrepancies.
- **10:** The facial features are perfectly matched to those in the reference images.

**Pay detailed attention to these facial elements:**
- **Eyes:** Shape, size, spacing, color, and distinctive characteristics of the eyes and eyebrows.
- **Nose:** Shape, size, width, bridge height, and nostril appearance.
- **Mouth:** Lip shape, fullness, width, and distinctive smile characteristics.
- **Facial structure:** Cheekbone prominence, jawline definition, chin shape, and forehead structure.
- **Skin features:** Distinctive marks like moles, freckles, wrinkles, and overall facial texture.
- **Proportions:** Overall facial symmetry and proportional relationships between features.

**Example:** If the instruction requests combining the face from one image onto another pose, the final image should clearly show the same facial features from the source image.

**Important:**
- For each significant facial feature difference, deduct at least one point.
- Ignore hairstyle, body shape, clothing, background, pose, or other non-facial elements.
- Focus only on facial similarity, not whether the overall instruction was followed.
- **Scoring should be strict**—high scores should only be given for very close facial matches.
- Consider the level of detail visible in the images when making your assessment.

**Editing instruction:** `<instruction>`

---

