# OpenReview forum: "WithAnyone: Toward Controllable and ID Consistent Image Generation"
_ICLR.cc/2026/Conference — ICLR 2026 Poster_

### Official Review · Reviewer_6GUJ · 2025-10-29

**Soundness:** 4
**Presentation:** 4
**Contribution:** 3
**Rating:** 6
**Confidence:** 5

**Summary:**

To address the conflict between identity consistency and controllability in text-to-image generation—where existing models tend to suffer from the "copy-paste" issue —the authors construct the large-scale MultiID-2M dataset, which contains 500k group photos with paired references and 1.5 million unpaired group photos. They also design MultiID-Bench, a benchmark for quantifying copy-paste artifacts and the trade-off between ID fidelity and variation. Additionally, the authors propose the WithAnyone model based on the FLUX architecture. By leveraging an ID contrastive loss (with an extended negative sample pool) and a ground-truth-aligned ID loss, combined with a four-stage training pipeline (reconstruction pre-training with fixed prompts, reconstruction pre-training with full captions, paired fine-tuning, and quality tuning), WithAnyone reduces copy-paste artifacts while maintaining high ID similarity. Experiments validate its advantages in pose/expression controllability and perceptual quality.

**Strengths:**

1. Solid experiments: A large number of data experiments and qualitative results support the observations presented in this paper.
2. Promising research direction: Most existing works on ID consistency focus on ID consistency itself, while systematic research on editability-related topics is lacking. This paper conducts in-depth exploration of this idea, proposing corresponding data-driven solutions and relevant benchmarks. I believe the research community should invest more efforts in this area to achieve a better trade-off between similarity and editability, rather than overemphasizing similarity.

**Weaknesses:**

1. Lack of innovation in the experimental design: Most of the methods adopted have been explored by previous studies (e.g., contrastive loss), or only involve minor modifications based on prior work (e.g., GT-Align).
2. The experimental results are suboptimal: The reduction in similarity is quite significant. For instance, the second ID from the right in the bottom image of Figure 12 does not closely resemble the reference identity.

**Questions:**

1. All cases in this paper are based on celebrity images. What are the results when applying the model to ordinary people or synthetic portraits?

---

> ### Author Response · Authors · 2025-11-19
> **Author Response to Reviewer 6GUJ**
>
> We sincerely appreciate the reviewer’s constructive and insightful feedback. Below we provide clarifications addressing the raised concerns.
>
>
>
> **📝 W1. Perceived Limited Innovation**
>
> **💡A:** We respectfully clarify that, although some components (e.g., contrastive loss) exist in prior work, our contributions are non-trivial and enabled by new settings unique to our study:
>
> - **A quantitative formulation of the copy-paste problem.**
>   As the reviewer noted, the field tends to overemphasize similarity, often producing near-copy-paste results. Our Multi-ID Bench introduces ground-truth target IDs, allowing—for the first time—a systematic and quantitative study of (i) similarity vs. copy-paste, (ii) the trade-off between them, and (iii) the distribution shift between real ID pairs and generated images.
>
> - **A large-scale paired dataset enabling new learning paradigms.**
>   To the best of our knowledge, *Multi-ID 2M* is the first publicly available, high-quality, large-scale paired dataset for ID-specific generation, as summarized in Tab.4. The proposed curation pipeline is scalable and reproducible, and it provides the data foundation required to study editability beyond pure similarity.
>
> - **A four-phase training strategy made possible by the new dataset.**
>   Our training framework is intentionally designed to exploit the paired structure of Multi-ID 2M. Each phase contributes distinct learning effects, as demonstrated in the ablation study (Tab.6). The full pipeline cannot be implemented without the dataset’s unique characteristics.
>
> - **ID Contrastive Loss with an extended negative pool.**
>   While contrastive learning is not new, no prior work has been able to construct a sufficiently large negative pool for ID-specific generation due to the lack of large paired datasets. The availability of Multi-ID 2M enables contrastive learning to operate effectively at scale and substantially improves editability.
>
>
>
> **📝 W2. Suboptimal Visual Resemblance in Some Cases**
>
> **💡A:** We acknowledge that individual examples may show varying resemblance levels, especially under challenging style-change requests. However, our quantitative evaluation and user study consistently demonstrate that **WithAnyone achieves state-of-the-art performance in both similarity and non-copy-paste generation**. The specific case highlighted by the reviewer represents an edge case but does not reflect the overall performance trends.
>
> ---
>
> **📝 Q1. Generalization to Non-celebrity Identities**
>
> **💡A:** OmniContext is a benchmark containing non-celebrity identities. Beyond the quantitative results provided in the main submission, we have added additional qualitative comparisons using non-celebrity references in the revised appendix (Fig. 16). These results show that our model generalizes well to ordinary individuals. We attribute this to (i) the diversity of our training data and (ii) the model’s explicit disentanglement of ID and editability.
>
> ---
>
> We thank the reviewer again for the valuable feedback and hope the above clarifications address the concerns.

---

### Official Review · Reviewer_VDLA · 2025-10-30

**Soundness:** 3
**Presentation:** 3
**Contribution:** 2
**Rating:** 6
**Confidence:** 4

**Summary:**

The paper studies identity-consistent image generation conditioned on a reference face image and a text prompt. The authors argue that current identity-preserving generators often fall into a "copy-paste" failure mode: instead of synthesizing a coherent new face, they directly paste the reference face into the generated scene, leading to visible artifacts in pose, expression, and lighting mismatch.

To address this, the authors introduce (1) a large-scale paired dataset containing multi-person images together with target face identities, aimed at training models that handle multi-person composition; and (2) a new evaluation benchmark with metrics designed to distinguish between naive face copy-paste and proper identity transfer, by explicitly comparing the generated face to both the provided reference and the intended target identity.

Building on these resources, the paper proposes a four-phase finetuning pipeline based on the Flux image generation model, following the PuLID architecture, to improve multi-identity conditioning and reduce copy-paste behavior. Empirically, the finetuned model WithAnyone is reported to perform competitively with SOTA approaches in both single-person and multi-person generation settings.

**Strengths:**

1. The paper makes a conceptual step forward by explicitly distinguishing between the reference face and the intended target face, moving toward more controllable identity-consistent person generation.

2. It contributes a large-scale paired dataset with diverse identities and multi-person compositions, which is likely to be valuable for future work on identity-conditioned generation.

3. It proposes new evaluation metrics designed to capture the copy-paste failure mode in identity-consistent generation, offering a more diagnostic way to assess whether a model is truly preserving identity rather than naively pasting faces.

4. The proposed model achieves strong results and is competitive with SOTA methods in both single-person and multi-person generation settings.

**Weaknesses:**

1. *Single-person quality not clearly superior*. In the single-subject setting, the qualitative comparisons do not convincingly demonstrate an advantage of WithAnyone. In particular, the claim that the method avoids copy–paste style artifacts would be more credible if shown under challenging conditions (extreme lighting, large pose changes, strong facial expressions). The revision should include more targeted visual examples in these stress cases.

2. *Underpowered user study*. The user study is based on only 10 raters, which limits statistical reliability and generalizability. A larger participant pool is needed, ideally with demographic diversity and significance reporting.

3. *Limited ablations*. The ablation analysis is incomplete. It would be valuable to (i) report performance after each training phase/stage to show incremental gains, and (ii) include a baseline trained with only the diffusion/denoising loss to isolate the benefit of the proposed objectives.

4. *Unclear ID contrastive loss formulation*. Equation 5 defines the ID Contrastive Loss using features from the reference image rather than the generated (target) image, but the paper does not justify this design choice or discuss alternatives. The rationale and expected effect on identity preservation need to be explained.

5. *Inconsistent table highlighting*. The visual highlighting in Tables 1 and 2 appears misleading. For example, in Table 1(a), Qwen-Image-Edit seems to have the best CP score among non-ground-truth methods, but WithAnyone is marked as “best.” The authors should fix or clarify the ranking/formatting so that highlighted cells actually correspond to the strongest scores.

**Questions:**

How Ground-truth-Aligned ID Loss is computed is still unclear to me. How do authors obtain the generated face image to compute this loss at high noise level? Is it to estimate an approximate $z_0$ or to run the whole denoising process to obtain a clean $z_0$? I would expect a more detailed explanation as the current description in the paper doesn't distinguish WithAnyone with existing methods in obtaining generated images.

---

> ### Author Response · Authors · 2025-11-19
> **Author Response to Reviewer VDLA**
>
> We sincerely thank the reviewer for the constructive and detailed feedback. The comments helped us identify issues in the presentation, especially regarding the ID Loss and ID Contrastive Loss, and we have substantially revised the manuscript to address these concerns.
>
>
> **📝 W1. Results under Challenging Conditions**
>
> **💡A:** We appreciate the suggestion to evaluate performance under more challenging conditions. In the revised manuscript (Fig. 15), we provide additional comparisons under stress conditions, including low resolution, motion blur, extreme lighting, and flare. These results demonstrate that *WithAnyone* maintains identity consistency without copy–paste artifacts even under difficult scenarios, surpassing prior methods that tend to replicate the artifacts present in low-quality reference images.
>
>
>
> **📝 W2. User Study Scale**
>
> **💡A:** We agree that a larger user study would strengthen the statistical reliability. However, each annotator must evaluate four dimensions across five methods for 230 samples to ensure consistency, which requires approximately four full days of work per annotator. Given the substantial manual effort involved, 10 raters represents the maximum feasible within our budget constraints.
>
>
>
> **📝 W3. Limited Ablations**
>
> **💡A:** We have expanded our ablation analysis in the revision, now including performance after each training phase (Table 6). We did not include an ablation without the ID Loss because, in practice, the model fails to converge meaningfully without it. The ID Loss is critical throughout training, and all our attempted variants without it produced degraded or unstable results.
>
>
>
> **📝 W4. Clarification on ID Contrastive Loss**
>
> **💡A:** We sincerely apologize for the confusion. You are absolutely correct: the positive sample is indeed the ground-truth target rather than the reference image. This was an error in the initial manuscript, but not in our practical implementation. We have corrected the formulation and added more explanations.
>
>
>
> **📝 W5. Table Highlighting**
>
> **💡A:** We apologize for the misleading presentation. As noted in the table captions, only methods with a Sim_GT above a specified threshold are ranked. When a method is not sufficiently similar to the ground truth, a high CP score does not indicate strong performance, as being dissimilar trivially avoids copy–paste artifacts. We have clarified the ranking scheme and updated the visual formatting to avoid confusion.
>
> ---
>
> **📝 Q1. Ground-truth-Aligned ID Loss Computation**
>
> **💡A:** We apologize for omitting this important detail due to page limitations. A full explanation is now included in Appendix E.1.
>
> In summary, for computing the Ground-truth-Aligned ID Loss from a noisy latent, we directly denoise the latent using the predicted velocity and decode it into pixel space. This produces a training-time estimate of the generated face (visualization at different timesteps can be found in Fig.7). This approach differs from prior works, and we discuss its advantages and disadvantages in the revision. Unfortunately, we cannot provide direct comparisons with previous implementations because no prior works have released their training pipelines.
>
> ---
>
> We again thank the reviewer for the insightful and technically grounded comments. We believe the revisions have significantly strengthened the clarity and completeness of our manuscript.

---

### Official Review · Reviewer_wdT7 · 2025-10-31

**Soundness:** 3
**Presentation:** 3
**Contribution:** 3
**Rating:** 6
**Confidence:** 4

**Summary:**

This paper terms the failure mode of the reconstruction-based diffusion training where it fails to produce diversity as **copy-paste**, such that it focuses too much on fidelity and cannot adapt to human preference and diversity requirement. This paper proposes a well annotated MultiID-2M benchmark to provide diverse references for each identity under multi-person scenarios, addressing balance between identity preservation and generation diversity with a contrastive identity loss (and corresponding metrics to measure performance).

**Strengths:**

1. There are previous benchmark papers which propose datasets of similar functionalities, but MultiID-2M is larger in scale (2M images) and uniquely focuses on group photos with labeled identities, beyond objects, anime characters, and human faces. The rich annotation also enables fine-grained control aspects. Thus, I think the high quality of MultiID-2M is the main strength/contribution of this paper and cannot be substituted by simply assembling a few previous datasets.

2. Using contrastive loss directly on reference images sounds clever to me. There are previous work which adds contrastive loss to diffusion model, e.g. Customcontrast [1] pulls representations of the same subject closer. WithAnyone links target, generated, and reference images together via reconstruction loss + contrastive loss, which repurposes contrastive learning in another way, but I think this paper is among the first papers to leverage contrastive learning here, which sounds interesting.

[1] Chen, Nan, et al. "Customcontrast: A multilevel contrastive perspective for subject-driven text-to-image customization." Proceedings of the AAAI Conference on Artificial Intelligence. Vol. 39. No. 2. 2025.

**Weaknesses:**

1. The intuition/motivation from the ''copy-paste'' effect has been proposed and studied [2,3], where they also leverage multiple images per identity to resolve overfitting and target leakage that leads to copy-paste effect. Then, it will be better for the authors to compare their datasets and proposed methods to demonstrate the improvements from this work -- I agree MultiID-2M is more comprehensive and unique as I summarized in strength 1, but it will be good to discuss it in the manuscript to better illustrate the motivation and improvement.

2. Following weakness 1, since it is a grounded problem with previous studies, this paper shares some similarities in their approach and evaluation. For example, [2,3] use multiple different images of the same identity as the target image, and [3] also proposes metrics to measure the alignment/balance between fidelity and diversity. This weakens the novelty of this work for the motivation part.

3. I hope there can be more ablation studies to demonstrate what makes this method works better, besides the good MultiID-2M data. A. I see ablation on number of negative examples for InfoNCE, but I think it is more of parameter tuning, can the authors show results w/o contrastive loss at all? B. Compared to previous work, is the capability of learning from MultiID reference images brought by larger models than initially SD 1.4/1.5? I am curious about what contributes the most to make WithAnyone works well on MultiID-2M, whether it is due to contrastive loss or larger model's expressivity. I suggest try baseline minus CL loss, baseline with reduced model size, and baseline minus CL loss and with reduced model size.

[2] Li, Zhen, et al. "Photomaker: Customizing realistic human photos via stacked id embedding." Proceedings of the IEEE/CVF conference on computer vision and pattern recognition. 2024.

[3] Tang, Haoran, et al. "Retrieving conditions from reference images for diffusion models." arXiv preprint arXiv:2312.02521 (2023).

**Questions:**

Please see Weaknesses for full details. In summary, weakness 1 and 2 need clarifications and updating manuscript to more precisely summarize the motivation and contribution of this paper, and weakness 3 needs additional experiments and explanations to help audience better understand the proposed method, e.g. which components/properties contribute the most. I am happy to raise my rating if my concerns are well addressed, especially for weakness 3, because I think such ablations provide new understanding and shed light on future work, adding strength to this paper.

---

> ### Author Response · Authors · 2025-11-19
> **Author Response to Reviewer wdT7**
>
> We sincerely thank the reviewer for their thoughtful and constructive feedback. We address the concerns point by point below.
>
> **📝 W1. Dataset Comparison with [2, 3]**
>
> **💡A:** Thank you for the suggestion. We have added the comparisons with the datasets in [2] and [3] in Tab. 4. In brief, the dataset in [2] includes only single-person paired data, and [3] focuses on anime-style faces. Our dataset is substantially more comprehensive: it contains both single-person and multi-person paired data, along with a high-quality subset and a stylized augmentation subset, providing broader coverage and stronger supervision. In addition, neither of the two prior datasets is publicly available, whereas we will release ours to support future community research.
>
> **📝 W2. "Copy-paste" Metric Comparison with [2, 3]**
>
> **💡A:** Prior works have noted the copy–paste phenomenon, but mostly through qualitative analysis rather than identity-sensitive metrics. LPIPS-based diversity in [2] measures deviation from the reference, while the CLIP-T metric in [3] captures high-level semantic changes but ignores fine-grained identity cues. Neither is designed for diagnosing ID-level copy–paste.
>
> Following [2], we compute LPIPS between each generated face and both its ground-truth (**LPIPS_GT**) and reference (**LPIPS_Ref**) on MultiID-Bench, and define an **overfit ratio**: the percentage of generated images more similar to the reference than to the ground truth.
>
> However, LPIPS behaves inconsistently with visual assessments: LPIPS_GT is often lower than LPIPS_Ref, and LPIPS-based overfit ratios show no stable trend, contradicting both qualitative observations and the near-100% overfit ratios measured by ArcFace.
>
> This exposes a core limitation of [2]’s metrics: while useful for semantic diversity, LPIPS fails to capture identity similarity and is unreliable for ID-specific copy–paste evaluation. In contrast, MultiID-Bench—with explicit ground-truth identities
>
> | Method | LPIPS_GT | LPIPS_Ref (PhotoMaker[2]↑) | Overfit Ratio LPIPS (%) | Overfit Ratio ArcFace (%)| Copy-Paste ↓|
> | --- | --- | --- | --- | --- | --- |
> | DreamO | 0.621 | 0.728 | 46.11 | 98.25 |  0.179 |
> | OmniGen | 0.565 | 0.562 | 45.88 | 99.42 |  0.209 |
> | OmniGen2 | 0.589 | 0.612 | 40.72 | 90.70 |  0.081 |
> | GPT | 0.609 | 0.524 | 83.33 | 81.71 | 0.061 |
> | UNO | 0.581 | 0.555 | 61.86 | 82.38 | 0.043 |
> | UMO | 0.578 | 0.548 | 63.40 | 98.45 |  0.176 |
> | UniPortrait | 0.563 | 0.538 | 63.92 | 98.45 |  0.254 |
> | ID-Patch | 0.598 | 0.625 | 36.69 | 99.42 | 0.183 |
> | Ours | 0.599 | 0.697 | 37.10 | 97.83 | 0.161 |
>
> **📝 W3.1 Ablation on Contrastive Loss**
>
> **💡A:** We apologize for the unclear labeling in Fig.15. The curve **"0 × InfoNCE"** represents training **without** the contrastive loss (Fig.15 in the initial version, Fig.17 in the revision). The extended negative pool is not just a tuning choice: adding 1000× more negatives per batch follows established principles that large negative pools substantially strengthen contrastive learning.
>
> **📝 W3.2 Comparison with Other Methods Using the Same Base Model**
>
> **💡A:** We agree that base model improvements can significantly affect downstream performance. WithAnyone surpasses many strong SOTA methods with the same `FLUX` backbone—e.g., `UNO`, `UMO`, `USO`, `PuLID`, `DreamO`. Hence, its gains cannot be solely attributed to larger models, as most baselines share comparable backbone capacity.
>
> **📝 W3.3 Ablation on Smaller Models**
>
> **💡A:** We agree that smaller-model ablations would provide further insight. However, a smaller version of `FLUX.1` is not publicly available. Ablations on `SD1.4/SD1.5/SDXL` require substantially different training recipes and tuning, and we are working to deliver results during the discussion period. We will provide updates once experiments are complete.
>
> ---
>
> **📝Q: What Contributes to the Improved Performance of WithAnyone**
>
> **💡A:** We summarize the key findings with quantitative evidence (Tab.6, revised manuscript)::
>
> - **Training Phases**
>   - *Phase 1:* establishes initial ID preservation; fixed prompts accelerate convergence.
>   - *Phase 2:* enhances ID consistency, reaching peak similarity and copy-paste scores.
>   - *Phase 3:* paired tuning significantly reduces copy-paste while maintaining or slightly improving Sim_GT; WithAnyone’s main reduction occurs here.
>   - *Phase 4:* retains quantitative performance while introducing style diversity.
> - **GT-aligned ID Loss:** stronger supervision during fine-tuning (Fig.7), improving ID similarity (Tab.3).
> - **ID Contrastive Loss**
>   - Without extended negatives: minimal effect (Fig.17).
>   - With extended negatives: faster convergence (Fig.15) and higher final performance (Tab.3).
>
> **Summary:** WithAnyone’s superior performance arises from
> 1) the large-scale, high-quality Multi-ID 2M dataset with paired training,
> 2) the critical role of the ID Loss for stable learning, and
> 3) additional benefits from ID Contrastive Loss with extended negatives.

---

> > ### Comment · Reviewer_wdT7 · 2025-11-28
> > **Thank you for your clarification**
> >
> > Dear authors,
> >
> > Thank you for your clarification and explanation that solves my concerns, I now better understand the improvement and difference from the previous work, and better read your ablation studies. Regarding ablation on model size (expressivity), I definitely agree that the gains cannot be solely attributed to larger models, and I understand that it may not be feasible to run ablation on new models during rebuttal phase. I was suggesting some way to strengthen the paper with a finer breakdown of the effectiveness of each modular design, but this study is solid itself. I will maintain my positive rating, thank you.

---

### Official Review · Reviewer_YRvY · 2025-11-02

**Soundness:** 3
**Presentation:** 3
**Contribution:** 3
**Rating:** 8
**Confidence:** 4

**Summary:**

This paper makes a strong and timely contribution to controllable, identity-consistent image generation by introducing WithAnyone, a diffusion-based framework trained with a novel contrastive identity loss and a ground-truth–aligned ID loss to mitigate “copy-paste” artifacts—where models overfit to reference faces instead of generalizing identity

**Strengths:**

The paper’s main strengths lie in its clear problem formulation, methodological thoroughness, and comprehensive evaluation. It identifies and formalizes the “copy-paste” artifact in identity-consistent image generation, offering a concrete metric to quantify this often-overlooked issue. The introduction of the MultiID-2M dataset and MultiID-Bench benchmark represents a valuable infrastructural contribution, enabling more systematic and reproducible research on multi-identity generation. The proposed WithAnyone framework integrates a ground-truth–aligned identity loss and a contrastive loss with extended negatives in a diffusion-based pipeline, leading to measurable improvements in controllability and identity preservation. Extensive experiments, strong ablation studies, and user evaluations convincingly support the claimed benefits, while the paper’s clear presentation, detailed appendix, and ethical considerations reflect a high level of polish and maturity in execution.

**Weaknesses:**

The proposed “copy-paste” failure mode, while intuitively appealing, overlaps conceptually with previously discussed overfitting and reconstruction issues in personalization models such as DreamBooth or PuLID, making the novelty somewhat incremental rather than fundamental.

The dataset MultiID-2M, though large, is primarily composed of celebrity data scraped from the web, raising concerns about bias, consent, and representativeness; furthermore, the lack of public access to the full dataset severely limits reproducibility and undermines the paper’s claim to openness. Evaluation metrics like the Copy-Paste score depend heavily on specific embedding spaces (e.g., ArcFace), which could bias results toward models aligned with that embedding and may not fully capture perceptual similarity.

The use of a similarity metric is clearly insufficient to justify identity consistency. Additional face verification metrics are needed, EER, FRM1000 etc.

**Questions:**

Generalization to Non-Celebrity or Low-Quality Data: Since MultiID-2M primarily contains celebrity faces with high-quality, well-lit imagery, how does WithAnyone perform on non-celebrity or in-the-wild datasets with lower resolution, occlusions, or varied lighting conditions? Would the model’s performance degrade significantly outside of this curated domain?

Dataset Ethics and Bias: Although the paper addresses ethical data sourcing and the exclusion of minors, could the authors elaborate on how potential biases (e.g., nationality, ethnicity, or gender imbalance) in MultiID-2M might impact the model’s fairness and generalization? Have any bias analyses or mitigation strategies been conducted?

Reproducibility and Accessibility: The paper states that only a small subset of MultiID-2M is included in the supplementary material. Are there concrete plans or timelines for releasing the full dataset and benchmark, or alternative means (e.g., data access agreements) to allow other researchers to replicate results?


Validity of the Copy-Paste Metric: The proposed Copy-Paste (MCP) score depends on cosine distances in face embedding space. How sensitive is this metric to the choice of embedding model (e.g., ArcFace vs. ElasticFace or CurricularFace etc)? What about face verfic?ation metrics, e.g., FMR10000? Have the authors verified that the metric aligns with perceptual similarity judgments beyond the limited user study?

**Details Of Ethics Concerns:**

While the paper includes an explicit ethics statement and claims that all data were collected from publicly available sources, it heavily relies on large-scale scraping of celebrity face images. This raises several ethical and legal concerns, including potential lack of informed consent from identifiable individuals, even if images are publicly accessible.

---

> ### Author Response · Authors · 2025-11-19
> **Author Response to Reviewer YRvY**
>
> We sincerely thank the reviewer for the constructive feedback and recognition of our contribution. Below we provide concise responses to the raised concerns.
>
> **📝 W1. Novelty of the Copy-Paste Failure Mode**
>
> **💡A:** While related issues like overfitting or unintended reconstruction have been mentioned in prior personalization works (e.g., `DreamBooth`, `PuLID`), our study *formally defines* and *quantifies* this failure mode, and introduces a benchmark with ground-truth identities to systematically analyze it — which is non-trivial.
>
> Our proposed MultiID-Bench allows controlled investigation of the trade-off between similarity and unintended copying, offering a complementary and meaningful contribution.
>
> Additionally, we conducted a new experiment using a metric similar to that in `PhotoMaker`, as described in our response to **wdT7**, further validating the utility of our proposed copy–paste metric. Please refer to **W2** for details.
>
>
> **📝 W2. Reproducibility and Dataset Accessibility**
>
> **💡A:** We understand the concerns regarding reproducibility. Upon acceptance, we will release all necessary components to fully reproduce our results, including:
> 1) **MultiID-Bench**: evaluation samples and runnable codebase;
> 2) **MultiID-2M**: training data and reference identity base;
> 3) **WithAnyone**: model family weights, inference code, and a Gradio demo.
>
> We also have drafted license and take-down policies to address potential misuse.
>
> **📝 W3. Embedding-Space Bias from Using ArcFace for Both Supervision and Evaluation**
>
> **💡A:** We agree that using the same embedding model for both training supervision and evaluation could introduce bias. To avoid this:
>
> - We use **antelopev2** for identity supervision, and **buffalo_l** (a different ArcFace model) for evaluation.
> - Additionally, we evaluate with **FaceNet** which is widely used in face similarity metric in face customization research community and with an extra **AdaFace** similarity. We report the average across three independent embeddings in the main paper.
>
> This demonstrates that our metric and improvements are *not* tied to a single embedding model.
>
>
> **📝 W4. Use of Similarity Metrics vs. Face-Verification Metrics**
>
> **💡A:** We appreciate the suggestion to include verification-oriented metrics such as EER or FMR10000. These metrics are standard for identity *verification*, but not directly suited to *customized image generation*, where defining “imposters” is non-trivial. Although none of the compared SOTA methods report such metrics, we agree that they could enhance the comprehensiveness of our benchmark.
>
> To address this, in MultiID-Bench we treat all other IDs in the test set as imposters and report FNMR (%) at FMR = 0.1% and 0.01% for further analysis.
>
> | Method | FMR1000 (GT)(\%) ↓| FMR1000 (Ref)(\%) ↓| FMR10000 (GT)(\%) ↓| FMR10000 (Ref)(\%) ↓| Copy-Paste ↓|
> |---|---|---|---|---| ---|
> | DreamO| *7.88* | 2.46| 19.21  | *4.43*| 0.179
> | OmniGen| 19.00| 4.00| 36.00 | 12.00   | 0.209
> | OmniGen2| 31.68  | 21.78| 56.44  | 32.18  | 0.081
> | GPT| 11.96 | 5.43     | 35.33 | 7.61 | 0.061
> | UNO| 52.74| 29.85  | 69.65| 47.76  | 0.043
> | UMO| 7.96| **1.00** | **11.94** | **1.99** | 0.176
> | UniPortrait  | ***1.58*** | ***0.53***|***7.37***| ***1.05*** | 0.254
> | ID-Patch | 11.05  | 2.91  | 26.16| 4.65|0.183
> | Ours    | **4.47** | *2.23*  | *12.85*   | 4.47 | 0.161
>
> ***First***, **Second**, *Third* are marked accordingly.
>
> Our method ranks second when compared to GT and third when compared to reference images. Notably, UniPortrait achieves the highest copy-paste score, giving it a strong advantage under binary metrics: by directly copying the reference identity, it yields artificially high similarity and lower error rates, without truly preserving identity semantics.
>
> ---
>
> **📝 Q1. Generalization to Non-Celebrity & Low-Quality Data**
> **💡A:** We added qualitative results in the appendix (Fig. 15–16), including low-quality references and non-celebrity identities from OmniContext. WithAnyone shows stable performance with only mild degradation under extreme quality issues, outperforming prior methods that even copy artifacts from low-quality references.
>
>
> **📝 Q2. Dataset Bias & Fairness**
> **💡A:** We acknowledge potential bias from large-scale web data. Within the limited work cycle, we have tried our best to balance the bias (e.g. manually incorporated ~400 additional African American identities ) and will continue expanding for better demographic balance.
>
>
> **📝 Q3. Reproducibility & Release**
> **💡A:** See **W2**
>
>
> **📝 Q4. Sensitivity to Embedding Choice**
> **💡A:** We tested the copy-paste metric using ArcFace-buffalo_l, AdaFace, and FaceNet. Rankings and conclusions stay consistent, indicating robustness to embedding choice. Detailed results are provided in Tab. 5. Although FaceNet yields more uniform scores, WithAnyone remains competitive, and user studies confirm strong alignment with perceptual judgments.

---

### Author Response · Authors · 2025-11-19
**Overall Response**

We express our sincere appreciation to all reviewers for their thoughtful and constructive feedback. We find most comments highly valuable and have revised the manuscript accordingly to improve clarity and completeness. Additional experiments and analyses have been conducted to address specific concerns raised by the reviewers.

We summarize the detailed difference in the revision:

1. **[Main text]** In **Eq.5**, the positive anchor is corrected to be the ground-truth target image instead of the reference image.
2. **[Appendix]** In **E.1**, we provide the missing explanation of how we acquire pixel-space estimates of generated images during training for computing the Ground-truth-Aligned ID Loss.
3. **[Appendix]** In **Fig.15**, we add more qualitative comparisons under challenging conditions (low resolution, motion blur, extreme lighting, flare).
4. **[Appendix]** In **Fig.16**, we add more qualitative comparisons using non-celebrity reference images from OmniContext.
5. **[Appendix]** In **Tab.5**, we add results of Identity Metrics evaluated by three different embedding models (ArcFace-buffalo_l, AdaFace, FaceNet). In the main text, the metrics are reported as the average across the three embeddings.
6. **[Appendix]** In **Tab.6**, we expand the ablation study to include performance after each training phase.
7. **[Appendix]** In **Fig.19**, we demonstrate the effectiveness of training phase 4.

---

### Author Response · Authors · 2025-11-26
**Follow-up on Overall Response**

Dear Reviewers,

Thank you all again for your constructive feedback on our submission.

We are writing to follow up on our response posted last week. We would greatly appreciate it if you could let us know if our rebuttal has sufficiently addressed your concerns.

We remain fully available to answer any additional questions or provide further clarifications during the remaining discussion period.

Best regards,

All authors

---

### Meta-Review · Program_Chairs · 2025-12-21

**Summary:**

Reviewers generally agreed that the paper tackles an important and well-motivated problem in identity-consistent image generation, namely the copy-paste failure mode caused by reconstruction-based training and similarity-driven evaluation. The proposed dataset (MultiID-2M), benchmark (MultiID-Bench), and extensive experiments were viewed as solid and carefully executed.
The main concern focused on method-level novelty. Several reviewers noted that the core components—paired identity training, contrastive identity losses, and face-embedding–based supervision—are conceptually related to prior work, raising questions about whether the improvements stem from fundamentally new techniques or from better data and training protocols. Additional concerns included the generality of the evaluation, particularly beyond celebrity identities, and the clarity of attribution of performance gains to individual components.
The rebuttal addressed many of these issues by adding ablations, stress tests, and clearer justification of the benchmark design, which improved confidence in the empirical results. Overall, remaining concerns are primarily about the degree of novelty rather than technical soundness or experimental validity.

**This paper is being conditionally accepted provided the authors address the following in their camera-ready:**
[Ethics concerns]: The authors should describe their data sources and detail the steps they have taken to get consent from the featured individuals. They should also note in their ethics statement the fact that this is dual-use technology, and the possible ethical harms that may arise from ID consistent image generation.
** Conditions for acceptance have been satisfied.**

**Reviewer Concerns:**

Concerns addressed by the rebuttal:
(1) The authors clarified the motivation and formulation of the copy-paste problem and justified the proposed benchmark design, emphasizing the distinction between similarity to reference images and similarity to ground-truth targets.
(2) Additional quantitative experiments, stress tests under challenging conditions, and expanded ablations improved confidence in the empirical robustness of the results.
(3) The rebuttal better explained the role of paired identity training, GT-aligned ID loss, and extended negative contrastive loss, helping attribute performance gains to specific components.

Concerns still outstanding:
(1) Questions regarding method-level novelty remain to some extent, as key components (e.g., contrastive identity supervision and face-embedding–based alignment) overlap conceptually with prior work, and the methodological contribution is largely incremental.
(2) The generalization of the approach beyond celebrity-based datasets and face-centric identity definitions is not fully resolved.
(3) While improved, the evaluation still primarily focuses on face similarity, and broader notions of identity consistency are not extensively explored.

**Reviewer Scores:**

Reviewer 1: Likely unchanged or slightly increased. This reviewer was generally positive about the problem formulation, dataset, and experimental rigor, and most of their concerns were adequately addressed in the rebuttal.

Reviewer 2: Likely slightly increased. The added ablations, clarifications on the copy-paste metric, and additional experiments appear to address the reviewer’s main technical questions, improving confidence in the empirical claims, though novelty concerns may remain.

Reviewer 3: Likely unchanged. While the rebuttal improved clarity and justification of the approach, the reviewer’s core concern regarding method-level novelty and overlap with prior work is only partially resolved, making a score change less likely.

---

### Decision · Program_Chairs · 2026-01-26

Accept (Poster)